# Rhodamine6G and Hœchst33342 narrow BmrA conformational spectrum for a more efficient use of ATP

A. Gobet [1,7], L. Moissonnier[2,7], E. Zarkadas [3], S. Magnard[2], E. Bettler[4], J. Martin [5], R. Terreux[4], G. Schoehn [6], C. Orelle [2], JM Jault [2], P. Falson [2] ✉ & V. Chaptal [2] ✉

Multidrug ABC transporters harness the energy of ATP binding and hydrolysis to translocate substrates out of the cell and detoxify them. While this involves a well-accepted alternating access mechanism, molecular details of this interplay are still elusive. Rhodamine6G binding on a catalytic inactive mutant of the homodimeric multidrug ABC transporter BmrA triggers a cooperative binding of ATP on the two identical nucleotide-binding-sites, otherwise michaelian. Here, we investigate this asymmetric behavior via a structural-enzymology approach, solving cryoEM structures of BmrA at defined ATP ratios, highlighting the plasticity of BmrA as it undergoes the transition from inward to outward facing conformations. Analysis of continuous heterogeneity within cryoEM data and structural dynamics, reveals that Rhodamine6G narrows the conformational spectrum explored by the nucleotide-binding domains. We observe the same behavior for the other drug Hœchst33342. Following on these findings, the effect of drug-binding showed an ATPase stimulation and a maximal transport activity of the wild-type protein at the concentration-range where the cooperative transition occurs. Altogether, these findings provide a description of the influence of drug binding on the ATP-binding sites through a change in conformational dynamics.

Drug resistance mediated by ABC (ATP-Binding Cassette) transporters contributes to the first line of defense for organisms, decreasing the intracellular drug concentration below cytotoxic levels[1–3]. A typical hallmark of multidrug ABC transporters is the ability to recognize and transport a wide array of structurally unrelated substrates across the plasma membrane, thereby protecting the cell against many xenobiotics. These transporters thus harbor a polyspecificity for multiple ligands, that they can accommodate within a rather large and adaptable binding pocket. The wealth of structural data available on these transporters as well as the biochemical and biophysical characterizations have highlighted their plasticity and their ability to adapt to various ligands[4–8]. At the core, it is clear that ABC transporters are very flexible and undergo significant conformational changes to perform their transport function.

Type IV ABC transporters involved in MultiDrug Resistance (MDR) phenotype perform their function by undergoing a series of

[1]Department of Molecular Biology and Genetics, Universitetsbyen 81, Aarhus C, Denmark. [2]Molecular Microbiology & Structural Biochemistry Unit. UMR5086 CNRS University Lyon-1. 7 passage du Vercors, Lyon, France. [3]Université Grenoble Alpes, CNRS, CEA, EMBL, ISBG, Grenoble, France. [4]ECMO team, Laboratoire de Biologie Tissulaire et d'Ingénierie (LBTI), UMR5305 CNRS University Lyon-1, 7 passage du Vercors, Lyon, France. [5]Laboratory of Biology and Modeling of the Cell, Ecole Normale Supérieure de Lyon, CNRS UMR 5239, Inserm U1293, University Claude Bernard Lyon 1, Lyon, France. [6]Université Grenoble Alpes, CNRS, CEA, IBS, Grenoble, France. [7]These authors contributed equally: A. Gobet, L. Moissonnier. ✉e-mail: pfalson@me.com; vincent.chaptal@ibcp.fr

conformational changes pertaining to the alternating access mechanism. They are formed of two transmembrane domains (TMD) forming a cavity at their center where substrates bind, and of two nucleotide-binding domains (NBD) binding and hydrolyzing ATP-Mg$^{2+}$ to fuel the system. MDR ABC transporters transport their substrates starting with an Inward-Facing (IF) conformation where substrates can bind. Binding of ATP-Mg$^{2+}$ to the NBDs bring them together, translating this conformational change to the TMDs which then open towards the outside (Outward-Facing, OF) where substrates are released. ATP hydrolysis and product release reset the transporter back to the IF conformation, ready for another cycle. While the overall scheme of this transport cycle is widely accepted, details of each step and their order are still highly debated.

The *Bacillus subtilis* efflux pump BmrA, belongs to the type-IV family of MDR ABC transporters[9–12], conferring resistance to cervimycin-C secreted by the biotope competitor *Streptomyces tendæ*[13]. A *B. subtilis* strain resistant to the antibiotic was isolated and contained mutations on the *bmra* promotor region increasing and stabilizing its mRNA, which results in BmrA overexpression at the membrane[12,13]. BmrA binds and transports many structurally-unrelated drugs, including doxorubicin or daunorubicin, Hœchst33342, Rhodamine6G (R6G) or ethidium bromide[10,12]. BmrA harbors a demonstrated plasticity to handle its ligands, thus conferring an evolutionary advantage to cells by detoxifying them against a wide range of xenobiotics. It has been postulated that instead of going through discrete well-defined states, it uses its intrinsic plasticity to swing back and forth around an intermediate occluded state while deforming to adapt to each transported molecule[9,12,14–17]. However so far, the molecular details of such critical plasticity and how it couples both functions of ATP hydrolysis and drug translocation lacks detailed structural information.

Here, we provide insights on BmrA plasticity undergoing IF to OF conformational changes and its ability to respond to ligand/substrate binding using a structural enzymology approach. We first probed ATP-Mg$^{2+}$ binding to BmrA in the absence or presence of R6G, showing a shift from michaelian to cooperative behavior. In order to get knowledge on what drives this change of behavior, we then visualized by cryoEM the effects of both ligand binding along this transition. We further analyzed protein flexibility within cryoEM data using a software that we developed[18], and reproduced these findings by Molecular Dynamics (MD) simulations. Data show that R6G focuses BmrA conformational changes, allowing for a more efficient space exploration. Furthermore, we show that binding of Hœchst33342 also changes BmrA's conformational space exploration, in a similar way to R6G but with its own amplitude highlighting the adaptation of BmrA to different ligands. Finally, we could establish that drug binding and shift in flexibility results in ATPase stimulation for ATP concentration where the transition occurs, resulting in a tighter coupling of transport and ATPase activities, otherwise uncoupled. Altogether, this study reveals how protein flexibility visualized by cryoEM data can explain the enzymologic behavior of BmrA.

## Results

### Shift from michaelian to cooperative ATP-Mg$^{2+}$ binding in response to R6G binding

BmrA is a dimer, each monomer bearing 1 TMD and 1 NBD[19]. The two NBDs are thus identical, and our recently solved OF structures of the E504A mutant confirmed that they bind ATP-Mg$^{2+}$ identically[12], in accordance with all other structures of type-IV ABC transporters. Probing ATP-Mg$^{2+}$ binding (Fig. 1A) indeed reveals a michaelian type of binding (hyperbolic-type curve, $K_{d\text{-app}} = 154.0\ \mu M \pm 49.0$). Importantly, the E504A catalytic mutant that still binds ATP but cannot hydrolyze it[20] was used in this study; this allowed to probe a unidirectional transition from IF to OF states, BmrA being blocked when reaching the OF conformation when complexed with ATP-Mg$^{2+}$ [9,12,14]. Interestingly,

addition of the substrate R6G prior to ATP-Mg$^{2+}$ binding resulted in a shift towards a sigmoidal-type curve, suggesting a cooperative binding of ATP-Mg$^{2+}$ (Fig. 1C), and implying that although identical the NBDs do not bind ATP-Mg$^{2+}$ the same way anymore ($K_{0.5} = 70.0\ \mu M \pm 2.6$, $h = 3.4 \pm 0.3$). Variation of intrinsic fluorescence is mostly due to changes in tryptophan residues environment, being the resultant of ATP-binding and conformational change. For BmrA E504A, the resting state without nucleotide is in the IF conformation (see below) and the ATP-bound form is in the OF conformation; conformational change is thus the main driver for a change in intrinsic fluorescence. This allows to assume that the pre-binding of R6G increased the apparent affinity for ATP, together with the steep transition suggesting that R6G binding ensures a more efficient conversion to the OF conformation mediated by ATP-Mg$^{2+}$ binding.

Since this behavior has consequences on the transport cycle, we performed a structural enzymology study to investigate this transition structurally. CryoEM grids were frozen at key ligand concentrations and imaged; without ATP-Mg$^{2+}$ in absence or presence of R6G (named E504A$^{apo}$ and E504A$^{R6G}$), at a 1:1 molar ratio ATP-Mg$^{2+}$:BmrA dimer (E504A$^{apo-25\mu MATP}$ and E504A$^{R6G-25\mu MATP}$), at the $K_{0.5}$ or close to the $K_{d\text{-app}}$ for ATP-Mg$^{2+}$ (E504A$^{apo-100\mu MATP}$ and E504A$^{R6G-70\mu MATP}$), and at saturating concentrations of ATP-Mg$^{2+}$ (E504A$^{apo-5mMATP}$ and E504A$^{R6G-5mMATP}$) (Fig. 1, Supplementary Figs. 1–8). Without nucleotide, BmrA is 100% in the IF conformation, in good agreement with SANS and HDX results[16], as well as cryoEM structures of the WT protein and A582C mutant[11] and single molecule FRET recent studies[21]. Increasing addition of ATP-Mg$^{2+}$ shifts the population towards the OF conformation to reach 100% of the population at saturating ATP concentrations. At a 1:1 molar ratio ATP-Mg$^{2+}$:BmrA in the presence of R6G, 25% of BmrA population already shifts towards the OF conformation. This is consistent with the cooperativity observed for ATP-Mg$^{2+}$ binding, and the notion that its binding to one Nucleotide-Binding Site (NBS) increases the affinity of ATP-Mg$^{2+}$ for the second NBS (Fig. 1D). Two molecules of ATP-Mg$^{2+}$ are thus trapped in the OF conformation, which add-up to trapping 50% of the ATP-Mg$^{2+}$ in the sample, and the clear shape of the OF conformation makes it distinguishable in the reconstructions. Not all BmrA shift to the OF conformation as the ATP-Mg$^{2+}$ concentration then drops below its apparent affinity, leaving the rest of the population in the IF conformation. In contrast, in the absence of R6G (E504A$^{apo-25\mu MATP}$), only IF reconstructions could be observed as ATP-Mg$^{2+}$ will distribute equally among both NBDs. No OF conformation is observed for this condition (Fig. 1B) despite the large number of particles picked on this particular grid, resulting in a high sampling of the different conformations; the OF conformation being very different from the IF ones, particle pertaining to that conformation are easily distinguishable and the absence of this observation is not due to a processing error but due to the absence of this conformation on the grid. The absence of OF conformation in the apo form compared to the R6G-bound one is probably due to the way the two proteins react in terms of intrinsic fluorescence. In presence of R6G, the shift in fluorescence in Fig. 1C is mostly dominated by conformational change as the population of conformations coincides well with the shape of the curve, while for the apo BmrA (Fig. 1A) the shift is most likely a mix between sensing ATP-binding and conformational change. At the $K_{0.5}$ for ATP-Mg$^{2+}$, 60% of the population is switched to the OF conformation in presence of R6G (E504A$^{R6G-70\mu MATP}$), while greater concentrations of ATP-Mg$^{2+}$ are required to reach a similar population distribution in the absence of drug. Altogether, this structural enzymology approach exemplifies how R6G binding on BmrA results in a more efficient conversion from IF to OF conformation, over a narrower range of ATP-Mg$^{2+}$ concentration.

### Flexibility of BmrA in the IF and OF conformations

3D models were built into each high-resolution 3D reconstruction to gain molecular insights on the IF to OF conversion (Fig. 2,

Supplementary Fig. 9). Clear electron densities could be observed for R6G in all the IF and OF reconstructions where it was added. In contrast, all reconstructions where R6G was absent did not show density in the substrate-binding cavity (Supplementary Fig. 10). For the OF conformation, current observations represent a biological duplicate of observations we previously made in ref. 12, with R6G wedged between TM1-2 and TM5′-6′ of each half transporter, with a total of 2 R6G per BmrA dimer (Fig. 2D, E). Binding of R6G triggers a closure of TM1 towards TM2, reinforcing the hand-fan motion of these 2 TMs previously hypothesized[12]. Admittedly, density for R6G is not very well defined, allowing for multiple positioning of the molecule, which denotes intrinsic R6G flexibility within the drug-binding pocket, as was also observed by molecular dynamics (MD) simulations[12]. In the IF conformations, R6G could also be modeled within the TMD in-between the same TM helices. Here again, electron density denotes multiple positions adopted by the molecule (also observed in MD simulations below), in line with the ability of BmrA to recognize structurally-unrelated ligands. To exemplify this, we have modeled R6G in 2 conformations of the benzoate moiety with 180° flip of the xanthene core fitting equally well (Fig. 2C, left). In E504A$^{apo}$, E504A$^{apo-25\mu MATP}$ and E504A$^{apo-100\mu MATP}$, multiple IF conformations of BmrA are observed with a rearrangement of TM helices and the NBD (Fig. 2A–C), in line with the flexibility observed for BmrA[11,16] and other ABC transporters in this conformation[4,6,22]. Notably, the E504A$^{apo}$ structure is exactly the same as the WT or A582C mutant structures recently solved (rmsd = 0.52 Å over 902 atoms and 0.45 Å over 883 atoms, respectively; Supplementary Fig. 11[11]). R6G binding in E504A$^{R6G}$ and E504A$^{R6G-25\mu MATP}$ results in a further rearrangement of TM helices within the dimer, with a wider opening of TM4-5-6′. Surprisingly, the rearrangement does not occur within a half-transporter, as it overlays well with BmrA in absence of R6G, but in the relative

orientation of the two halves (Fig. 2C, right part of the structure on which the overlay is performed). The effect of R6G binding further translates all the way to the NBD, resulting in a reorientation of the 2 NBD that face each other more directly (Fig. 2B), hinting at an influence on ATP binding observed in Fig. 1. Altogether, these structures highlight the flexibility of BmrA in the IF conformation, and the fact that R6G has an allosteric effect that reorients the protein at the ATP-binding site level, reminiscent of what was observed for ABCB1 in presence of drug or inhibitor[23].

## Impact of R6G on BmrA dynamics and conformational space exploration assessed by variability analysis of cryoEM data

With cryoEM structure resolution, it is possible to visualize inter-particle difference within a reconstruction, allowing to deconvolute protein movements in several directions of latent space[24]. Note that these changes can be observed through different methods available in many software, nicely reviewed in ref. 25. Importantly, the movements that can be observed result from different interpretation of the latent space and on how to deconvolute the movement. Software using neural networks to deconvolute movements, such as mannifoldEM, 3DFlex, CryoDRGN[26–28] or others use non-linear interpretation of the data to show deformations linking sub-states within the particle stack. These methods are very powerful to highlight special deformations undergone by the protein within each sub-state but will not model the link between different sub-states. In a different manner, linear models of conformational heterogeneity, such as multibody refinement or 3DVA[24,29], perform a principal component analysis of particle variance compared to the consensus reconstruction. This later analysis allows to decode general types of movements present within the particle stack, decomposing complex movements into simplified motions; the global movements undergone by the protein are thus visualized

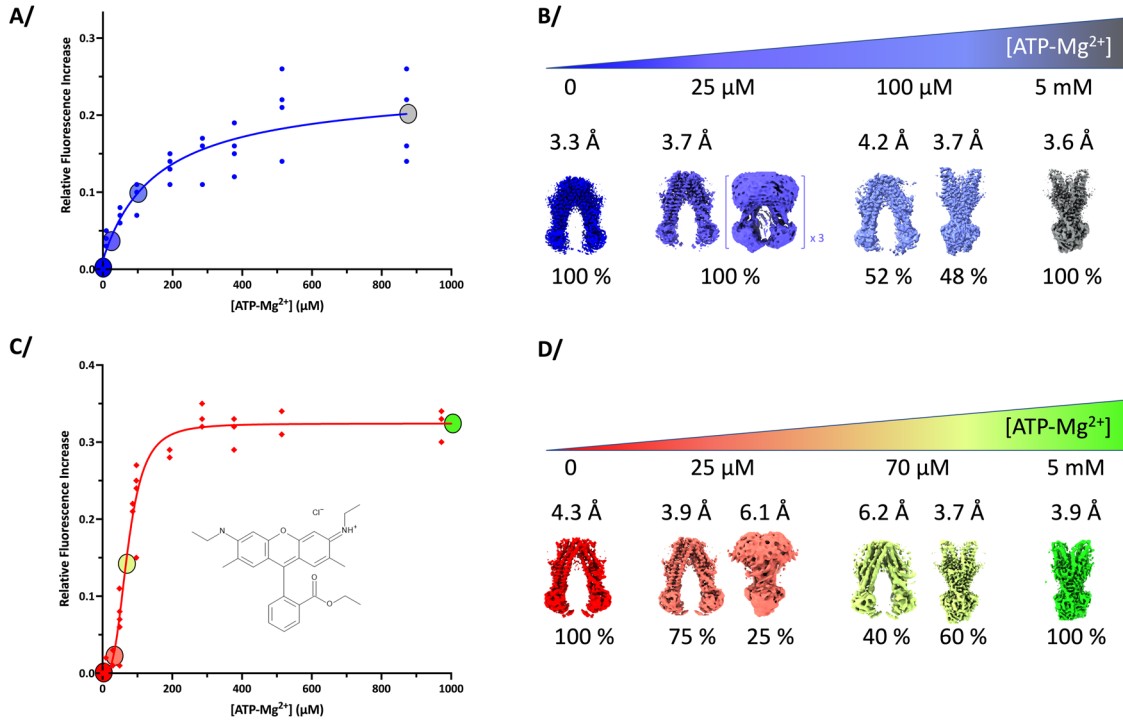

**Fig. 1 | Binding of ATP-Mg²⁺ on BmrA E504A in absence or presence of R6G and corresponding maps. A** Binding curves of ATP-Mg²⁺ measured by relative intrinsic fluorescence increase in the absence of Rhodamine 6G is michaelian. Circles correspond to the conditions at which cryo-EM data were collected. Blue to gray circles correspond to different condition of collected cryo-EM data at 0 μM, 25 μM, 100 μM and 5 mM ATP-Mg²⁺ for 25 μM of BmrA dimer in the sample. **B** Coulomb potential maps corresponding to each condition of collect and their relative proportions. For E504A$^{apo-25\mu MATP}$, one reconstruction could yield high resolution and 3 additional volumes could be resolved to IF conformations. **C** Binding curve of ATP-Mg²⁺ in presence of 100 μM R6G is cooperative. Cryo-EM data were collected at different concentration of ATP-Mg²⁺. Red to green circles correspond to the four conditions: 0 μM, 25 μM, 70 μM, and 5 mM ATP-Mg²⁺. **D** Coulomb potential maps corresponding to each condition and their relative proportions.

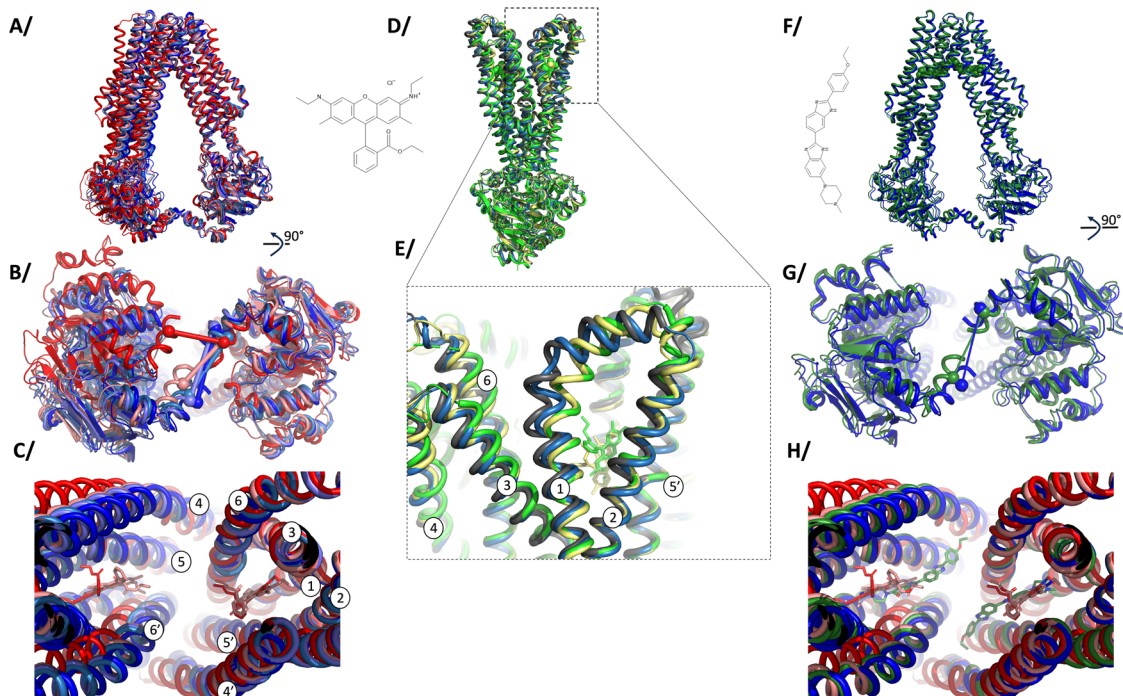

**Fig. 2 | Superposition of models in IF or OF conformations.** Colors match the ones of Fig. 1 and Supplementary Fig. 9. Protein in cartoon and ligands are in sticks, TM helices are numbered when appropriate. **A**−**C** BmrA in the IF conformation seen from the side, from under the NBD or a close-up view of the substrate-binding pocket, respectively (E504A$^{apo}$ in blue, E504A$^{R6G}$ in red, E504A$^{25\mu MATP}$ in skyblue, E504A$^{100\mu MATP}$ in lightblue, E504A$^{R6G-25\mu MATP}$ in salmon). Overlay was performed on the 3 first TM helices of chain A, residues 1-161. On (**B**), a line is drawn between residues 577 of each monomer to highlight the conformational change. **D**, **E** show BmrA in the OF conformation (E504A$^{100\mu MATP}$ in lightblue, E504A$^{5mMATP}$ in gray, E504A$^{R6G-70\mu MATP}$ in yellow, E504A$^{R6G-5mMATP}$ in green), with (**E**) being a close-up view of the TM1-2 loop with R6G in its pocket. Overlay was performed on the last 4 helices of chain A, residues 104-300 as in ref. 12. **F**−**H** BmrA in the IF conformation in complex with Hœchst33342 (E504A$^{H33342}$) seen from the side, from under the NBD or a close-up view of the substrate-binding pocket, respectively. E504A$^{H33342}$ is colored in green cartoon and Hœchst33342 is shown as spheres in (**F**) and as sticks in (**H**) colored by atom type with carbons in green. E504A$^{H33342}$ is overlaid with E504A$^{apo}$ (blue) in residues 1-161 of chain A in (**F**−**H**), and shown with the structures complexed with R6G (E504A$^{R6G}$ in red) in (**H**) to asses ligand positioning. Chemical structures of R6G and Hœchst33342 are shown in the picture.

through several main components, allowing for a deconvolution of a complex movement into simplified sub-movements. This later analysis was chosen to analyze the current datasets as it can reveal the overall influence of R6G on BmrA deformations and conformational space exploration. 3DVA was computed in simple mode, resulting in adding weights to the consensus map in latent space directions calculated by principal component analysis, or intermediate mode calculating real maps along the same direction. In this case, both methods gave the same result.

Such calculation was applied to E504A$^{apo}$ and E504A$^{R6G}$ to visualize movements within BmrA as a response to R6G binding (Supp-movies). Of note, the particle stacks of these two datasets show similar distribution of particles in 3D (no special orientation of the protein within the ice), and a gaussian distribution in latent space, allowing to compare similar types of particle distribution, making the movements directly comparable (Supplementary Fig. 12). To fully understand underlying movements, we designed a software that refines an ensemble of models inside this ensemble of maps[30] and applied the procedure to BmrA (Supplementary Fig. 13). Each component shows different movements, with clear rotations and translations of the NBD, that originate from the kinks in TM helices (Fig. 3A, Supplementary Figs. 14–16). For E504A$^{apo}$, three types of movements could be inferred. First a 12° rotation of the NBD centered on the beta sheet (V363), is observed in the first component (component 0). Second, a 16° rotation of the NBD alone centered on E453 is observed, without movement of TM helices (Component 1). Third, an overall oscillation of the structure is observed with translations moving the NBDs 5 Å away from each other (component 2). In presence of R6G (E504A$^{R6G}$) similar protein motions are observed and can thus be directly compared to E504A$^{apo}$;

the principal components in which these movements appear are different as expected for this type of analysis. The first NBD rotation centered on V363 is conserved in E504A$^{R6G}$, while the rotation centered on E453 is reduced by more than half (Fig. 3B). Notably, the oscillation that bring the NBDs closer together is greatly affected as the E504A$^{R6G}$ structure is stiffened and movements are very reduced (less than 1 Å) (Fig. 3A, B). This analysis shows that 1/ the NBDs explore a large conformational space by undergoing rotations and translations. The presence of R6G influences one rotation by reducing half of its amplitude; importantly the NBD movements start from a different point in presence of R6G with the NBDs already oriented closer to one another. 2/ The presence of R6G stabilizes the NBDs slightly further apart with a reduced translation towards each other. Overall, these motions show that R6G decreases the space exploration of the NBDs. At first, this interpretation might seem counter-intuitive as we would expect NBDs to close more easily according to the cooperativity observed for ATP-Mg$^{2+}$ binding, but one has to keep in mind the different time-scale of the experiment. By combining Molecular Dynamics simulations with enzymology (cf. below), this observation of reduction of conformational space exploration makes sense and allows to propose a model for the allostery between drug-binding site and ATP-Mg$^{2+}$ binding and hydrolysis.

## Molecular dynamics simulations of E504A$^{apo}$ and E504A$^{R6G}$ in the IF conformation

We then subjected both proteins to MD simulations in a lipid bilayer to investigate movements in the IF conformation linked to R6G binding (Fig. 4, Supplementary Figs. 17–23. All analyses start at 0 for the production run). ATP-Mg$^{2+}$ was added to the walker A motif of each NBD to

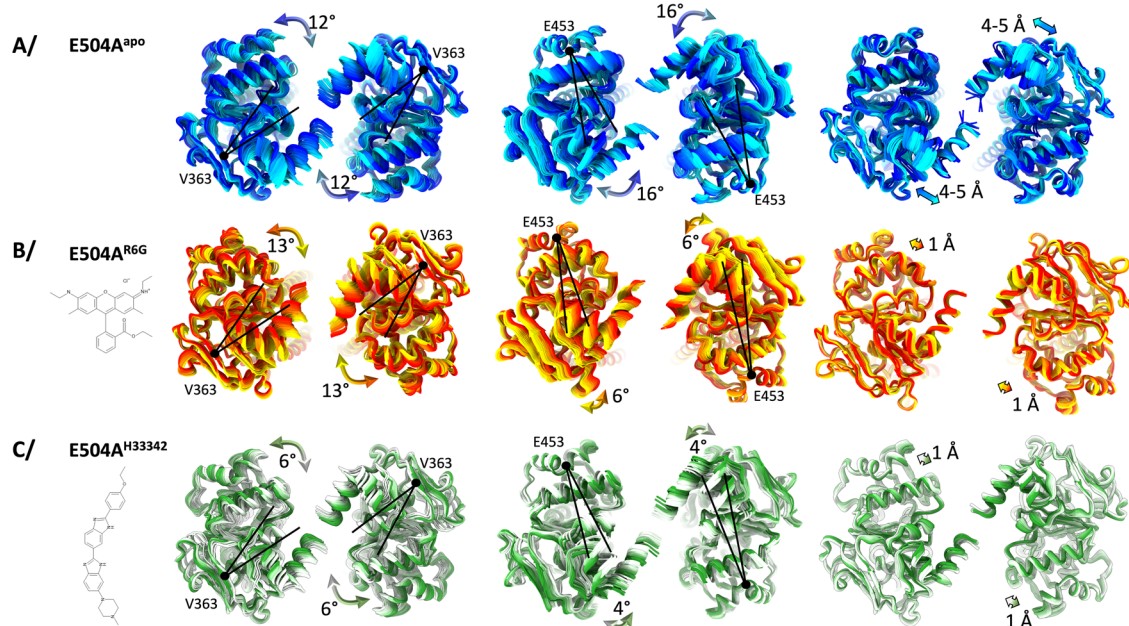

**Fig. 3 | Variability analysis of BmrA in the IF conformation with several ligands.** **A** 3DVA analysis of E504A[apo] in several components, resulting in 20 maps each. Models built by variability refinement[18] in the maps are represented in cartoon and colored from blue to cyan, with view from the NBDs. The main movement is represented on the structure with black lines representing the NBD rotation and the colored arrows depict the movement and its amplitude. Details of this analysis are shown in Supplementary Fig. 14. **B** same as A/ for E504A[R6G], colored from red to yellow. **C** same as (**A**) for E504A[H33342], colored from green to white.

incite closure of the NBDs during the simulations, which was observed. All replicates show BmrA movements towards a closure of the drug-binding cavity and the NBDs getting closer to each other (Fig. 4C, D). The amplitudes of the movements are in a different range in each replicate, but overall, all the distances aggregated show that both E504A[apo] and E504A[R6G] are able to move with similar range and fluctuate similarly (Supplementary Figs. 17–20); the drug thus does not prevent BmrA to move (Fig. 4E). However, the conformational space explored by E504A[R6G] is much more focused compared to E504A[apo] in which the NBDs explore a larger space (Supplementary Fig. 18), hinting at the allosteric influence of R6G on the NBDs. Along these lines, the NBD closures in E504[apo] and E504[R6G] are different. For E504[apo], closure happens in a concerted manner with the two NBDs getting closer to each other in a similar way. It results in a similar formation of both Nucleotide Binding Sites (NBS) as measured by the distances between the Walker A and signature C motif of the other monomer (Fig. 5A–C). This type of concerted closing is reminiscent to the tweezers-type of closing hypothesized for WT BmrA[11]. For E504A[R6G] however, closure of the NBS is asymmetric with the narrowing of one NBS more pronounced than the other one (Fig. 5D–F, the closing NBS is shown towards the top in Fig. 5F). For the first two replicates, the ATP molecule stays in close proximity of its canonical binding site, making that the simulation represents the closing of the first nucleotide-binding site when influenced by R6G binding in the trans-membrane domains. Simulations also reveal that R6G is mobile within its binding pocket but only explores the nearby space around its original position, owing to the hydrophobicity of the substrate-binding pocket (Supplementary Fig. 23). When R6G is present, TM helices mostly stay near their original positions, suggesting that R6G prevents direct closure of helices due to steric hindrance, while a stronger closure is observed in its absence (Fig. 4C, D). These observations correlate with the inter-particle difference in cryoEM data showing the effect of R6G-binding on NBD closure with a reduction and focusing of the conformational space explored by BmrA in the IF conformation and notably a reduction of direct closure.

## Impact of Hœchst33342 on BmrA dynamics and conformational space exploration

We investigated ATP-Mg$^{2+}$ binding when E504A is complexed to other ligands to probe if the transporter responds similarly as when complexed with R6G. Hœchst33342 and Doxorubicin bind to E504A (9.6 μM ± 1.3; 7.3 μM ± 2.2 respectively; Supplementary Fig. 24), with similar affinities as those observed for WT BmrA[10]. Interestingly, all ligands induce a cooperative ATP-Mg$^{2+}$ binding like R6G, with a similar $K_{0.5}$ (56.0 μM ± 2.6 $h = 2.0 ± 0.3$ for Hœchst33342; 65.5 μM ± 3.6 $h = 2.3 ± 0.3$ for Doxorubicin; Supplementary Fig. 24), suggesting a similar influence of ligands on ATP-Mg$^{2+}$ binding, albeit the varying hill slope between ligands suggests that a ligand-specific effect might occur. We solved the cryoEM structure of E504 in complex with Hœchst33342 in absence of ATP-Mg$^{2+}$ (named E504A[H33342], Fig. 2F–H, Supplementary Figs. 25–26). Here again, E504A[H33342] is 100% in the IF conformation. The high-resolution reconstruction differs slightly from E504A[apo], with a reorientation of the second half of the transporter induced by Hœchst-binding, leading to a further separation of the trans-membrane helices and a re-orientation of the second NBD (Fig. 2F, G). These modifications match the ones induced by R6G binding, with E504A[H33342] structure being very similar to E504A[R6G-25μMATP] (Fig. 2H). Two molecules of Hœchst33342 bind per BmrA dimer (Fig. 2F–H, Supplementary Fig. 26B), in a similar location as the one occupied by R6G. While Hœchst33342 and R6G share a part of the substrate-binding pocket, half of both Hœchst33342 molecules protrude out the binding-pocket to wedge between TM4,5 and 6, and TM4',5' and 6' respectively.

E504A[H33342] conformational space exploration was then studied using inter-particle variability of cryoEM data, in a similar fashion as performed for E504A[apo] and E504A[R6G]. Importantly, particle distribution and latent space particle distribution are similar across the 3 conditions studied (Supplementary Figs. 27 vs 12), granting the comparison of their movements through similar use of 3D variability analysis. As seen on Fig. 3C and Supplementary Fig. 28, Hœchst33342 has a greater impact than R6G on the same type of movements. Rotation of

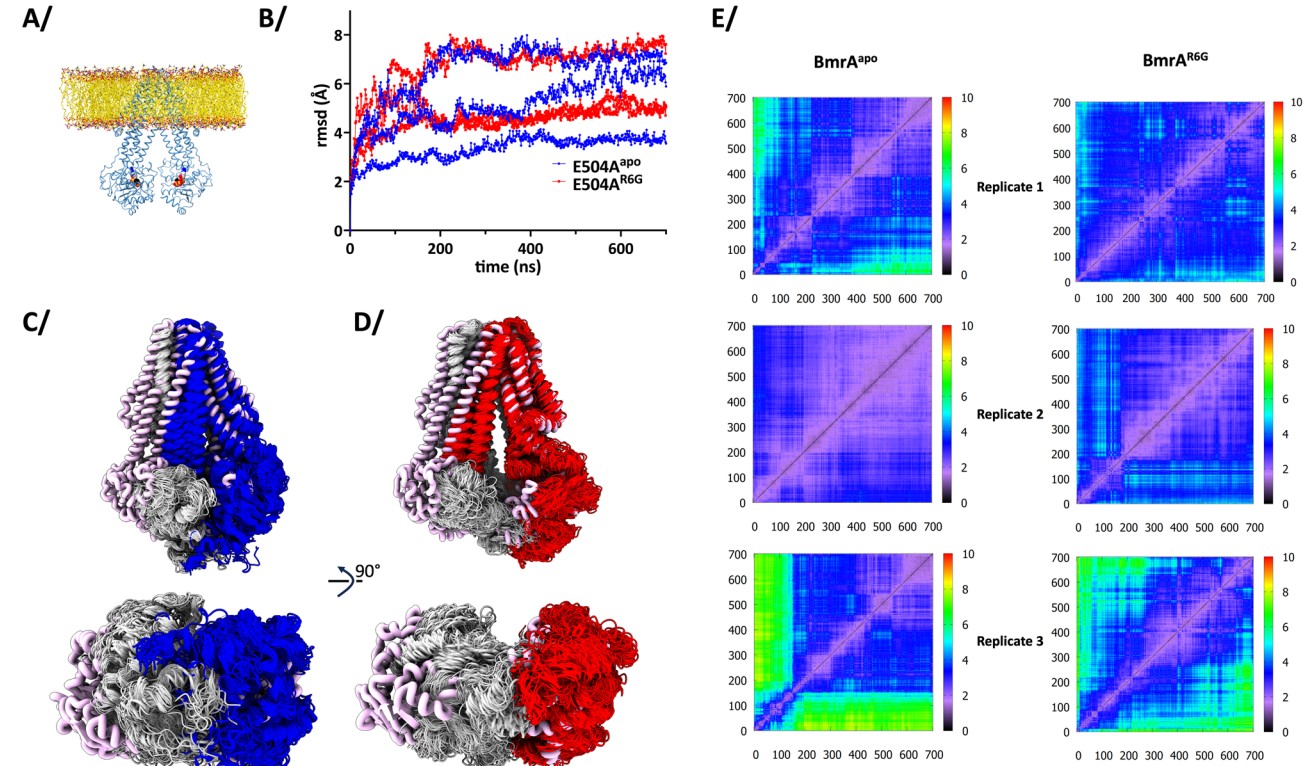

**Fig. 4 | Molecular dynamics simulations of BmrA in the IF conformation.**
**A** BmrA E504A was inserted in a lipid bilayer and ATP-Mg$^{2+}$ was added to start MD simulation, with or without R6G. **B** Rmsd of the MD trajectories for 3 replicates of the production run. The color code corresponds to (**C**, **D**). **C** Overlay of 3 replications of MD simulations of E504A$^{apo}$ to show the conformations sampled during the simulations. Thick pink ribbons correspond to the initial structure. **D** same as (**C**) for E504A$^{R6G}$. **E** 2D rmsd plots across the 700 ns long of simulations. Three replicates were made for each BmrA form E504A$^{apo}$ and E504A$^{R6G}$. X and Y axes are the frames. Color code on the right for the global rmsd between each frame, from low in black to high in red.

---

the NBD centered on V363 is reduced by half with only 6° sampled; rotation centered on E453 is impacted in a similar range as R6G with 4° sampled; finally, closure of the NBDs towards each other is abolished by Hœchst33342 binding, with 1 Å fluctuations observed corresponding to natural protein vibrations. These observations confirm that ligand-binding on BmrA impacts its conformational space exploration and result in a reduction of the space sampled by the NBDs.

## Impact of ligands on ATPase and transport activities

Since R6G induces a sharp transition from the IF to the OF conformation over a narrow range of ATP-Mg$^{2+}$ concentration, the effect on ATPase activity was investigated in this range. Indeed, when investigated at high ATP-Mg$^{2+}$ concentration with R6G, Doxorubicin or Hœchst33342, no stimulation was observed[10,31,32]. However, at low ATP-Mg$^{2+}$ concentrations, a clear stimulation is observed (Fig. 6A–C), corresponding to the concentration range where the IF to OF transition occurs. Similar stimulation is observed in detergent, liposome or in membrane vesicles (Fig. 6D). In all cases, where the ATP-Mg$^{2+}$ concentration is above the $K_{d-app}$ for ATP, i.e., ATP binding is not limiting anymore, no stimulation is observed confirming previous studies. We then investigated the transport efficiency for Hœchst33342 and Doxorubicin at varying ATP-Mg$^{2+}$ concentrations (Fig. 6E, F, Supplementary Figs. 29–30) as unfortunately R6G transport cannot be followed in membrane vesicles[12] but all three drugs nevertheless induce cooperative binding of ATP-Mg$^{2+}$ (Supplementary Fig. 24). We quantified on the same membrane fraction the Hœchst33342 transport and ATPase activity of BmrA, to determine the amount of substrate being transported in relation to the ATP consumption. The experiments were carried out the same day for a precise quantification of both activities and conducted in quadruplicates from two separate batches of membrane preparation. Hœchst33342 apparent transport per ATP hydrolyzed increases up to a maximum of 0.8 for 200–300 μM ATP-Mg$^{2+}$, and slowly decreases afterwards. This biphasic curve implies that two phenomena are occurring. At low ATP-Mg$^{2+}$ concentrations, a near strict coupling of Hœchst33342 transport with ATP hydrolysis happens, whereas ATP gets hydrolyzed faster than its use for substrate transport at higher ATP-Mg$^{2+}$ concentrations. For Doxorubicin, a similar curve is observed with a wider peak spreading over 300–1000 μM ATP-Mg$^{2+}$, and reaching an apparent maximum of 0.2 Doxorubicin transported per ATP hydrolyzed. This seems to also indicate that when ATP-Mg$^{2+}$ concentrations are low resulting in slow binding to BmrA, the drug is transported more efficiently in regard to ATP consumption, while this effect disappears when ATP concentrations rise.

## Discussion

Communication between the substrate-binding site and the NBDs in ABC transporters has been described for many transporters and in various settings (a few examples here[6,14,33–38]). It remains however difficult to study and to investigate at the molecular level being the result of multiple factors. Here we used the sharp effect produced by R6G on BmrA, to visualize the IF to OF transition, over a narrow range of ATP-Mg$^{2+}$ concentration; the use of the ATPase inactive mutant E504A, which still binds ATP-Mg$^{2+}$ but does not hydrolyze it, permits this structural enzymology approach as the mutant only undergoes the forward direction and is blocked in the OF conformation (Fig. 1). The allostery manifests in several ways, visible at the NBD site. First, there is a structural rearrangement of the NBDs towards each other. This is possible with the intrinsic plasticity of BmrA in the IF conformation that allows to sample different conformations (Fig. 2A–C), as also

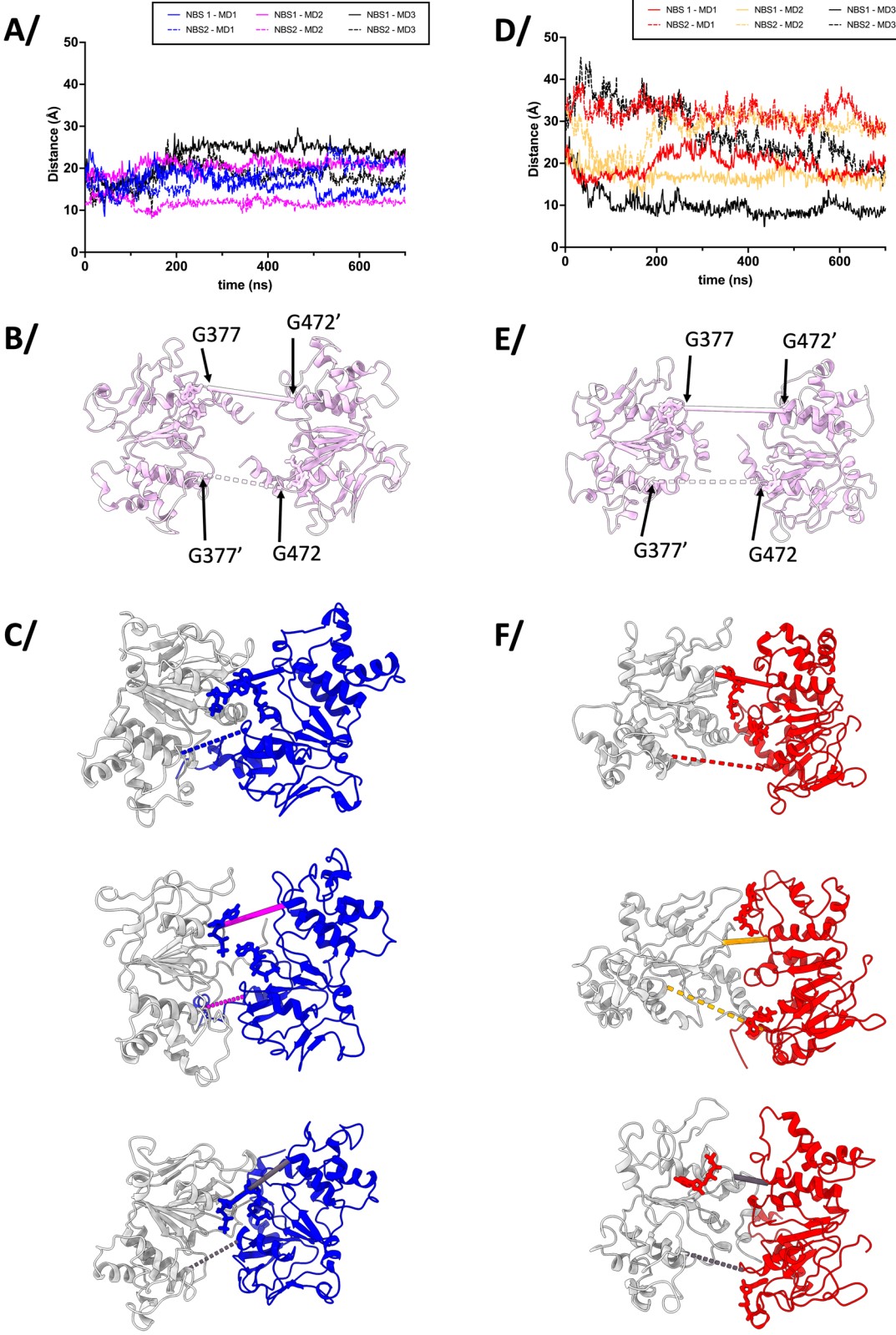

previously observed by NMR spectroscopy, HDX-MS or SANS studies[15–17]. Substrate binding to the drug-binding cavity thus optimizes the positioning of the NBDs. Second, the conformational space exploration of the NBDs is also influenced by R6G binding, as probed by the continuous heterogeneity analysis of cryoEM data (Fig. 3). R6G reduces the overall motions undergone by the NBDs, decreasing rotations and translations of the sites towards one another. This

reduction in space exploration is also visualized by MD simulations where R6G also focuses the motions sampled by the NBDs (Supplementary Fig. 18). In both cases, the NBDs are still very mobile, the presence of the drug does not prevent movement but rather allosterically influences the directions of the space exploration. We validated this observation by solving the cryoEM structure of BmrA in presence of Hœchst33342, a ligand that also induces cooperativity for

**Fig. 5 | NBS formation during Molecular Dynamics simulations. A** measure of the distance between the Nucleotide Binding Sites (NBS) during MD simulations for E504A$^{apo}$. The distances are measured between the Walker A motif of one monomer (G377) and the signature C motif of the other monomer (G472'), represented as solid line for 1 NBS and as dotted line for the other NBS. Replicate 1 is shown in blue (solid and dotted lines), replicate 2 in magenta and replicate 3 in black. **B** Representation of the NBDs seen from the trans-membrane domains of E504A$^{apo}$ for the cryoEM structure (initial structure), shown in pink cartoon with the ATP as

sticks. **C** The last frame of the MD simulation is shown in blue for monomer A and gray for the other monomer. The distances measured in (**A**) are shown as solid and dotted lines to match (**A**) with the same colors per replicate. **D** same as (**A**) for E504A$^{R6G}$ colored in red for replicate 1, orange for replicate 2 and black for replicate 3. **E** E504A$^{R6G}$ cryoEM structure (initial starting structure) and distances shown in pink as in (**B**). **F** same representation as (**C**) with monomer A colored in red. The NBS between G377 and G472 shown towards the top of the structures displayed is closing more than the one towards the bottom.

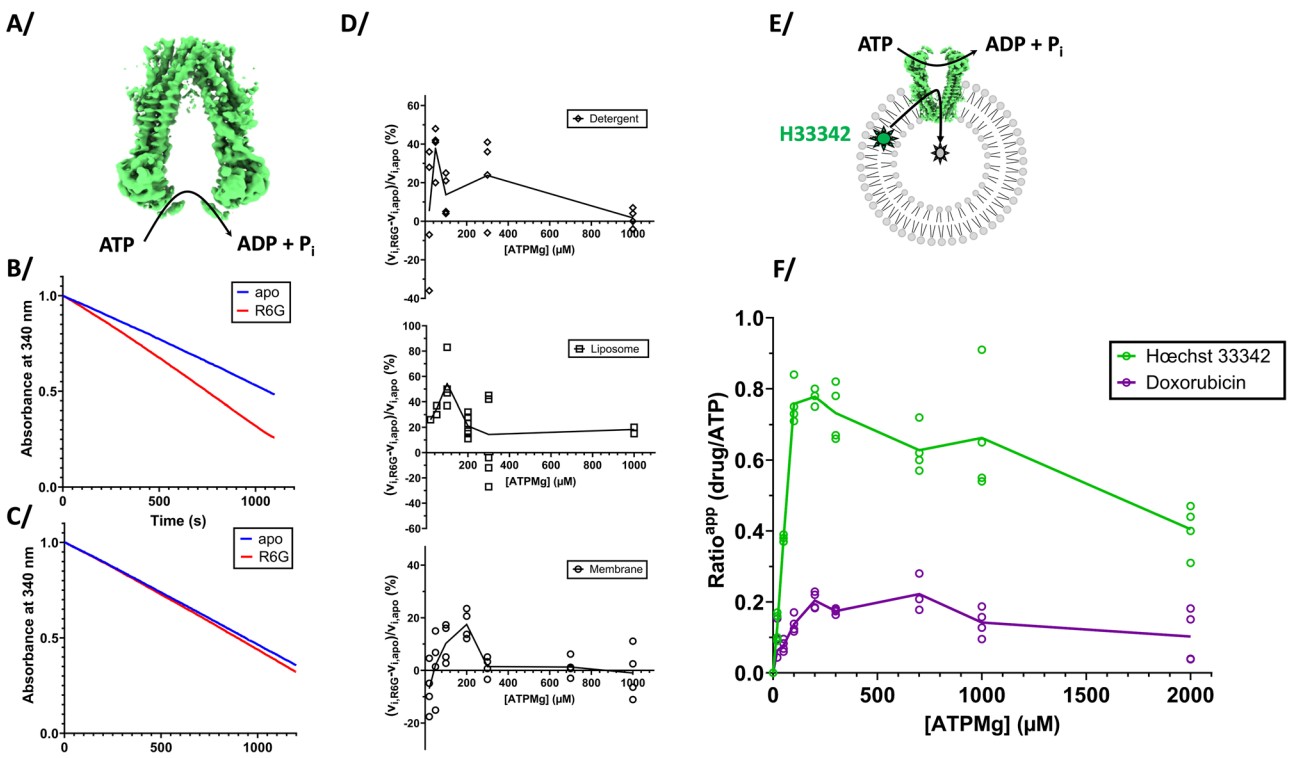

**Fig. 6 | ATPase activities and transport measurements. A** Schematic ATPase activity of the transporter **B** ATPase activity can be measured by following disappearance of ATP via a coupled enzymatic assay; decrease of absorbance at 340 nm is directly correlated to ATP hydrolysis. Blue curve for BmrA apo, and red in complex with R6G. The slope is greater with R6G showing a faster ATP hydrolysis. This example is for the point 300 μM ATP for BmrA in detergent as shown in (**D**). **C** same as (**B**) for 1 mM ATP showing no stimulation. **D** ATPase activity measured in detergent, liposomes or in membranes. Each point represents a kinetic experiment

as in (**B**) for a range of ATP concentration, acquired with or without R6G. The slope of each curve was measured at initial speeds, and the normalized ratio is presented on the graph to show the activation. **E** Scheme of Hœchst33342 transport in membranes occurring with ATP hydrolysis. **F** Transport efficiency of Hœchst33342 or Doxorubicin transported per ATP hydrolyzed along the range of ATP-Mg$^{2+}$ sampled in (**D**). Individual measures are indicated on the graph (quadruplicates) and the line links the average of each concentration of ATP-Mg$^{2+}$ sampled. Details of this experiment are shown in Supplementary Figs. 29–30.

ATP-binding with a similar apparent affinity and different Hill coefficient. Ligand-binding on BmrA was similar within the drug-binding pocket, but extruding more within the TM helices for Hœchst33342, its structure being longer than R6G. Hœchst33342 also induces a reduction in the conformational space exploration of the NBDs, confirming the observations made on R6G, and showing a further decrease in amplitude compared to R6G (Fig. 3). These are the first direct structural evidence of the influence of ligands on protein dynamics through space exploration.

Since the ATP-binding sites (NBS) are shared between the two NBDs, ATP-binding implies that the NBDs come in close proximity. However, R6G and Hœchst33342 prevent direct closure of the transporter using symmetrical closure as hypothesized for the apo WT transporter in absence of drug[11]; indeed, R6G and Hœchst33342 binding in the drug-binding pockets prevents the TM helices to deform and to directly come close to the symmetric mates in the dimer. We hypothesize that this forces the TM helices to deform around the drug, which will consequently form a first NBS with ATP sandwiched

between the Walker A motif of one NBD and the signature C motif of the opposite NBD (Fig. 7). This site would correspond to the high affinity site and would help prime the other NBS for ATP binding, in accordance with the cooperativity model. This hypothesis is visualized in two replicas of the MD simulations carried out on E504A$^{R6G}$ where we observe the closing of one NBS (Fig. 5D–F), supporting the claims made from all these observations. Of note, asymmetry at the NBS level during ATP hydrolysis has previously been observed by solid-state NMR for BmrA, illustrating that the two NBS can behave differently from each other thereby reinforcing the plastic behavior of this transporter[10,15].

Since the allosteric effect of R6G is to reduce the conformational space and to focus the movements explored by the NBDs, with a consequence of a sharp transition from IF to OF over a narrow range at ATP-Mg$^{2+}$, we hypothesized that the drug would stimulate ATP hydrolysis in this narrow range of ATP-Mg$^{2+}$ concentration. Even though the structures were obtained in a DDM/Cholate detergent mixture while the activities were performed on membranes, the two

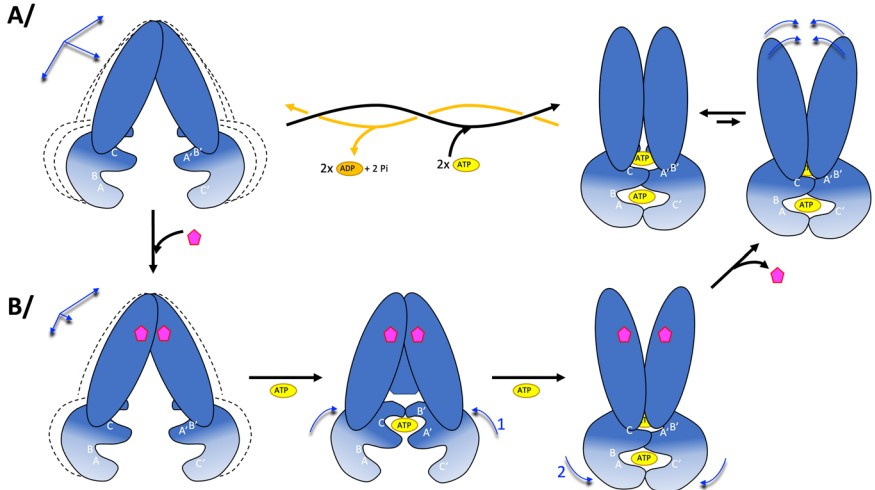

**Fig. 7 | Model of transport mechanism. A** In absence of substrate, BmrA WT in the IF conformation is very plastic. The 2 NBS bind 2 ATP-Mg²⁺ (yellow oval) identically, leading to the OF conformation. Plastic deformation is transferred to the other side of BmrA for release of a potential substrate; return to the IF conformation is achieved via ATP hydrolysis and release of ADP-Mg²⁺ (orange oval). The white letters A,B,C,A',B',C' refer to the Walker A, Walker B and signature C motifs of ABC transporters in the NBD. A NBS is formed by binding ATP between notably the A and B motifs of one NBD, and the C motif of the other NBD. **B** In presence of substrate, flexibility of the IF conformation is reduced and symmetrical closing is prevented by the presence of the drug. Binding of the first ATP-Mg²⁺ occurs by deformation around the drug, leading to the formation of the first NBS, which increases affinity for the other NBS and formation of the OF conformation. Using plasticity of the OF side of BmrA, substrate is released and the initial pathway of ATP hydrolysis is used to reset to the IF conformation.

environments are fairly close from BmrA perspective. We previously showed that the ATPase activities are the same between the two environments, as well as the binding affinities[12]. Also, the stimulation of ATPase activity was observed in detergent, liposomes or membrane vesicles, with a peak of stimulation centered on 100–300 μM ATP-Mg²⁺, similar for all environments and supporting the proposed model. This stimulation remains modest for BmrA compared to what can be observed for other Type-IV ABC transporters[39]. The maximum stimulation stays in the 30–40% activation, like what was also observed for reserpine, another substrate of BmrA[10]. This led us to investigate the relative stoichiometry of transport for BmrA of two substrates, Hœchst33342 and doxorubicin. They both induce cooperative ATP-Mg²⁺ binding like R6G (Supplementary Fig. 24) and their transport assays can be followed by fluorescence using inverted membrane vesicles[10,12,40]. For both substrates, a peak of substrate transported per ATP hydrolyzed is observed for the same range of 100–300 μM ATP-Mg²⁺ where stimulation occurs, as well as the transition IF to OF is maximal following ATP-Mg²⁺ binding. The ATP range for which the substrate stimulates ATP hydrolysis thus also corresponds to a more efficient drug transport (Fig. 6D–F). One can also note that Hœchst33342 and Doxorubicin are possibly transported with a different ratio of ATP usage by BmrA. Hœchst33342 and R6G binding pockets overlap in BmrA, like was also observed in the homolog human ABCB1[41]. Since there are 2 R6G and 2 Hœchst33342 bound on BmrA it is most probable that 2 Doxorubicin also bind in BmrA substrate cavity (Supplementary Fig. 24). Since 2 ATP-Mg²⁺ are required to achieve the OF conformation and the ratio Hœchst33342:ATP-hydrolyzed reaches values close to 1:1, it is likely that at the peak, 2 Hœchst33342 will be transported per cycle, resulting in an almost fully coupled activity. The smaller than 1 ratio also indicates that ATPase activity nevertheless occurs without any substrate bound to the drug-binding cavity or that the IF to OF transition occurs with only one Hœchst33342 bound. Also, at higher ATP-Mg²⁺ concentrations, ATP-Mg²⁺ binds faster and gets hydrolyzed faster than Hœchst is being transported, pointing rather to an uncoupling of the activities. Altogether, this would point to the diffusion of substrate inside the drug-binding pocket as a limiting factor for transport, as previously hypothesized by the kinetic selection model[42]. This hypothesis is reinforced with doxorubicin having an apparent much lower ratio towards ATP. Since the LogP of these two drugs are different (3 for Hœchst33342, 0.8 for Doxorubicin as determined by ACD percepta), it implies a lower partition of Doxorubicin within the lipid bilayer or lower accessibility from the cytoplasm to reach the drug-binding site. Altogether, this clearly reveals a decoupling between the activities of drug transport and ATP hydrolysis for BmrA. We thus hypothesize that the diffusion rate of drugs into BmrA binding pockets is a limiting factor for transport, while ATP is being used constantly[39]; if a drug is bound while a cycle is being performed upon ATP-Mg²⁺ binding, it will be transported.

The allosteric effect of drugs on ATP-Mg²⁺ binding (Supplementary Fig. 24) is nevertheless meaningful at low ATP-Mg²⁺ concentrations as it results in a stimulation of drug transport efficiency (Fig. 6F). Low ATP concentrations can be found in bacteria during early or stationary phase in rich or minimum media[43], starvation conditions[44], during sporulation or within persister cells[45]. In these conditions, ATP is scarce and precious, and cannot be wasted on a transporter working constantly; thus, drug binding would have a positive influence on ATP-Mg²⁺ binding and will help the transporter to transport them more efficiently by more strictly coupling the two activities, thereby protecting the cells against xenobiotics. Under more favorable growing conditions, ATP concentrations rise and the need for an allosteric effect is diminished as the transporter functions faster[39].

Finally, we observe here that R6G and Hœchst33342 modulate protein dynamics and conformational space exploration. This observation opens avenues of investigations to understand the structure and function relationship within proteins.

## Methods

### BmrA expression and purification

C43(DE3)ΔAcrB *E.coli* strain is transformed with the plasmid pET15b encoding for the BmrAE504A mutant fused to a 6-histidine tag at the N-terminal. A transformed colony is incubated in 3 mL of LB media supplemented with 50 μg/mL of ampicillin for 7 h at 37 °C with shaking. 30 μL from this day culture are diluted in 1 L of LB media containing 50 μg/mL of ampicillin and incubated overnight at 22 °C with agitation. When OD⁶⁰⁰ reaches 0.6, BmrA overexpression is induced by adding 0.7 mM IPTG. The culture is then incubated for 5 h at 22 °C under shaking. Bacteria are collected by centrifugation at 5000 × *g* for 15 min, 4 °C. The pellet is then suspended in 20 mL of 50 mM Tris-HCl pH8.0,

5 mM MgCl$_2$. Bacteria are lyzed by 3 passages with a disruptor Constant CellD system at 1.5 kbar, 4 °C. The bacterial suspension is centrifuged at 15,000 × $g$, 30 min, 4 °C. The supernatant is centrifuged for 1 h, 180,000 × $g$, 4 °C to pellet membranes. The membrane fraction is suspended in 25 mL of 50 mM Tris-HCl pH 8.0, 1 mM EDTA, anti-protease CLAPA 1X and centrifuged again with the same parameters. The membranes are finally suspended in 20 mM Tris-HCl pH 8.0, 300 mM sucrose, 1 mM EDTA, frozen in liquid nitrogen and stored at −70 °C.

Membranes are solubilized at 3 mg/mL in 20 mM Tris-HCl pH 8.0, 100 mM NaCl, 15% glycerol (v/v), anti-protease CLAPA 1X, 4.5% (v/v) Triton X100 and incubated 1 h and 20 min at 4 °C under gentle agitation. The solution is centrifuged 40 min at 100,000 × $g$, 4 °C. The supernatant is loaded onto a Ni$^{2+}$-NTA column pre-equilibrated with 20 mM Tris-HCl pH 8.0, 100 mM NaCl, 15% (v/v) glycerol, anti-protease CLAPA 1X, 4.5% Triton X100, and 20 mM imidazole. Resin is washed with 20 mM HEPES-NaOH pH 8.0, 100 mM NaCl, 20 mM imidazole, 1.3 mM DDM and 1 mM sodium cholate. Protein is eluted with the same buffer with 200 mM imidazole. Fractions of BmrA are pooled and diluted ten times in the same buffer as previously but without imidazole. The column is equilibrated with 20 mM HEPES-NaOH pH 8.0, 100 mM NaCl, 20 mM imidazole, 1.3 mM DDM and 1 mM sodium cholate, and the protein solution is loaded again for a second affinity chromatography. After elution, fractions of BmrA are pooled (typically 10 mL) and concentrated on 50 kDa cutoff Amicon Ultra-15 device at 1000 × $g$, 4 °C, until the volume reaches 500 μL. The solution is then injected on a Superdex 200 10/300 column (Cytiva) equilibrated with 20 mM HEPES-NaOH pH 7.5, 100 mM NaCl, 0.7 mM DDM and 0.7 mM sodium cholate.

## CryoEM grids preparation and data collection

Purified BmrAE504A was concentrated to 4 mg/mL as described above. Defined concentrations of ATP-Mg$^{2+}$ (25 μM, 70 μM, 100 μM, 5 mM final concentrations) are added to a final concentration of BmrA of 25 μM to be applied on the cryoEM grid. For the grid containing Rhodamine6G (R6G), the ligand is also added to a final concentration of 100 μM followed by a 15 min. incubation on ice prior to ATP-Mg$^{2+}$ addition. The mix was incubated 30 min. at room temperature to reach steady state before application on the cryoEM grid.

Ultra-Au 1.2/1.3 grids (Quantifoil) are glow discharged on air for 45 s at 30 mA (Emitech Glow Discharge). A volume of 3.5 μl of the mix is applied on freshly glow discharged grids at 20 °C and 100% humidity using a Vitrobot Mark IV (Thermofischer). Excess liquid is blotted 4 s at blot force 0 and 0.5 s drain time before vitrification in liquid ethane.

Data were collected on 2 Titan Krios at 300 kV equipped with a K3 direct electron detector (ESRF CM01 or Diamond eBIC), or a Talos Glacios equipped with a K2 detector (IBS, Grenoble). Data collection parameters are summarized in Supp Table 1.

## CryoEM data processing

Data were processed using cryosparc v2, v3 and v4 (Structura Bio) over the duration of the project, with the first data collected in Sept. 2019. Movies were submitted to patch motion correction and patch CTF estimation jobs, and particle picking was performed using blob picker (100–200 Å diameter) on a small subset of movies (usually 200) to create initial 2D classes to be used for template picking. Automatic particle picking was then performed on all the movies using the template from previous 2D classification. Mild 2D classification was operated to remove obvious bad particles (ice contamination or detergent micelles) but care was taken to not throw out "broken particles" at this stage. 3D maps are created with ab-initio reconstruction job asking for 7 models to observe the whole sample heterogeneity. The number of particles was noted for IF and OF classes to calculate their ratio (Fig. 1). Each map was then individually further refined using several rounds of hetero-refinement and non-uniform refinement until the highest

resolution was reached. Many routes were explored to reach high resolution reconstructions, leading to the final one displayed. Each job was computed with or without C2 symmetry to evaluate the impact on the maps; this is especially important for the datasets containing R6G to ascertain its density. In some cases, particle expansion followed by local refinement was performed to enhance the quality of the maps at the R6G binding site without imposing symmetry. The general data processing pipe was similar but some specificities were used for some datasets, to reach the best results possible. Summaries of data processing are shown in Supplementary Figs. 1–8. Notably, for the data set E504A$^{apo-100\mu MATP}$ an additional local CTF refinement was performed and a mask without detergent belt was applied. For data set E504A$^{apo-25\mu MATP}$ a mask without detergent belt and one NBD was used. For datasets showing discrete heterogeneity but where one conformation couldn't be resolved to high resolution, the reconstructions clearly show in which category (IF or OF) the transporter belongs (E504A$^{apo-25\mu MATP}$, E504A$^{R6G-25\mu MATP}$, E504A$^{R6G-70\mu MATP}$). These reconstructions were used to count the number of particles and thus derive percentages of populations, but no atomistic model was built.

## Model building

For the outward facing conformations, models were built using the previous OF structures of BmrA in absence or presence of R6G (PDB: 6R72 and 6R81[12]). Several cycles of manual building in Coot and real-space refinement using phenix.refine were performed. Final models were checked for rotamer and Ramachandran outliers.

For IF conformations, the first reconstruction was achieved before the opening of the AlphaFold database. The model was built using the OF conformation, with manual deformation of trans-membrane helices to match the density, using coot and ISOLDE, and several rounds of refinement in Phenix until a model was correctly built. Shortly after, the AlphaFold database was released and the BmrA model, which was generated in IF conformation, was also used for comparison and model building. It turns out that the conformation created by Alpha-Fold is more closed compared to the IF reconstruction experimentally obtained, thus distortions of that model was also needed. It was also used for comparison with our model building to ascertain the registry, which was identical in both models (ours and AlphaFold), and comforted us in residue assignment. For the other models in IF conformation, local modifications were performed by cycles of manual building and refinement as described before. As discussed below, the NBDs undergo significant movement in the IF conformation, and thus lacks clear electron density for the outmost regions. The core of the NBDs however is clearly defined, with key residues around W413 for example well defined and unambiguous side-chain assignment and positioning. For the outmost parts, the NBDs were placed following the clear model in the OF conformation. Final models and maps were deposited in the PDB and EMDB under the accession code listed in Supp-Table1.

## Variability analysis and variability refinement

For E504A$^{apo}$ and E504A$^{R6G}$, 3D Variability Analysis (3DVA, Cryosparc) was performed on the particles from the last non-uniform refinement. Resolution was filtered at 6 Å for 3DVA calculation and display for E504A$^{R6G}$, and at 4.5 Å for E504A$^{apo}$. Masks from Non-uniform refinement were initially dilated by 20 Å. The 3D variability display job was performed in simple or intermediate modes with similar results. 5 components were asked as output but the main movements are observed in the 3 first components, which were subjected to variability refinement. Each 20 maps originating from 3DVA were input to variability refinement (phenix.varref[18]) along with the final refined model. 50 models were generated for each map, with restraints adapted to the resolution of 3DVA calculations, and only the best model was kept for each map. The resulting file is a model file containing 20 models with atomic coordinates fitted to each map, allowing

for a visualization of the model in ChimeraX for detailed analysis. Movies were generated with ChimeraX.

## Construction of E504$^{apo}$ and E504$^{R6G}$ models embedded in lipid bilayer membranes

CHARMM-GUI[46,47] input generator was used to embed the different E504A$^{apo}$ and E504$^{R6G}$ models into a lipid bilayer composed by 1,2-dioleoyl-sn- glycero-3-phosphoethanolamine and 1,2-dioleoyl-sn-glycero-3-phospho-rac-1-glycerol with a respective ratio of 3:1. The original total size of every system was ca. $140 \times 140 \times 160$ Å$^3$ (see Supplementary Table 2 for system descriptions). To mimic physiological conditions, 0.15 M NaCl was used, and the systems were solvated using TIP3P explicit water molecules[48–50]. The final systems are made of ca. 264,329 atoms for E504$^{apo}$ and 268,405 E504$^{R6G}$ (see details in Supplementary Table 2).

## Molecular dynamics simulations

Rhodamine6 (R6G) substrate parameters were derived from the Generalized Amber Force Field version 2 (GAFF2)[51] using the Antechamber software[52]. R6G partial atomic charges were computed using AM1-BCC[53] semi empirical quantum mechanical based calculations.

CHARMM-GUI[46,47] outputs were in PDB format and were imported in MOE. For each condition E504A$^{apo}$ and E504A$^{R6G}$, ATP-Mg$^{2+}$ was added on the NBD in its binding-site by superposing the Outward-Facing conformation of BmrA (PDB: 6R82). After building protein-lipid systems; neutrality was ensured by randomly removing the corresponding number of counterions with the tleap software. Amber FF14SB[54], Lipid21[55] and the modified DNA.OL15[56,57] force fields were used to respectively model protein residues, lipids and ATP molecules. Water molecules, Mg$^{2+}$ ions and counterions were modeled using the TIP3P water model[48–50] as well as the corresponding monovalent and divalent ion parameters from Joung and Cheatham[58,59]. Each system was simulated with periodic boundary conditions. The cutoff for non-bonded interactions was 10 Å for both Coulomb and van der Waals potentials. Long-range electrostatic interactions were computed using the particle mesh Ewald method[60].

Minimization and thermalization of the systems and MD simulations were carried out with Amber22[61] using CPU and GPU PMEMD versions. Minimization was carried out in three steps by sequentially minimizing: (i) water O-atoms (2000 steps); (ii) all bonds involving H-atoms (5000 steps); (iii) whole system (5000 steps). Each system was then thermalized in two steps: (i) water molecules were thermalized to 100 K during 50 ps under (N,P,T) ensemble conditions using a 2 fs time integration in semi-isotropic conditions using Berendsen barostat; (ii) the whole system was then thermalized from 100 K to 300 K during 100 ps under (N,P,T) ensemble conditions with same condition for barostat. A pre-production step was introduced with semi-isotropic scaling barostat at 300 K using the production parameters for 100 ns. During this step we monitored that periodic boundary conditions parameters reached equilibrium. During this phase, the membrane was kept in contact with the protein and with the limit of the periodic boundary conditions. This pre-production phase was not included in the analyses. Production runs were then carried out for 700 nanoseconds with a triplicate starting from the same final equilibrated pre-production state. The production run with 2 fs integration timestep under (N,P,T) ensemble conditions with semi-isotropic scaling. Temperature was maintained using the Andersen-like thermostat[62]. Constant pressure set at 1 bar was maintained with semi-isotropic pressure scaling using Berendsen barostat[63].

Snapshots were saved every 200 ps. For each system, three replicas were performed to better sample the local conformational space. Each production run was carried out for 700 ns in triplicate, respectively for E504$^{apo}$ and E504$^{R6G}$. Only production runs were shown in the manuscript.

The PCA analysis was performed using the CPPTRAJ software[64] from the Amber-22 package on the dynamics previously performed. The protocol used was largely inspired by https://amberhub.chpc.utah.edu/introduction-to-principal-component-analysis/. Coordinate projections were visualized using VMD and the 'nmwiz' plugin[65].

## ATP binding

ATP-Mg$^{2+}$ binding was carried out by probing the intrinsic fluorescence change as a function of ATP-Mg$^{2+}$ concentration. Fluorescence was recorded on a SAFAS Xenius spectrophotofluorimeter set up at a constant photo multiplicator voltage of 570 V at 0.5 μM BmrA dimer. Tryptophan residues or N-acetyl tryptophan amide (NATA) used as negative control were excited at 290 nm, and their fluorescence emission spectra were recorded between 310 and 380 nm, with a 5-nm bandwidth for excitation and emission. NATA was used at the same concentration than that of BmrA tryptophan residues, typically around 6 μM. Experiments were carried out in a quartz cuvette in a final volume of 200 μL, in which increasing amounts of ATP-Mg$^{2+}$ were added. Stock solutions of 200 μM of ATP and MgCl$_2$ were individually made in the same buffer as the protein and buffered to pH 7.5, then mixed together in equal volume to make a 100 μM ATP-Mg$^{2+}$ initial stock solution. This mix was diluted to various stock concentrations, of which 2 μL were added to the cuvette, making the final concentrations tested. Resulting emission curves were integrated and subtracted from the same experiments carried out with the buffer without protein or NATA. Relative fluorescence changes were then calculated using the value of fluorescence at the emission peak (typically 325 nm for BmrA and 350 nm for NATA) and the equation $(F/Fo)/(F_{NATA}/Fo_{NATA})$ where F is the protein fluorescence at the given ligand concentration, Fo is the initial protein fluorescence without ligand. Same for $F_{NATA}$ and $Fo_{NATA}$. $F_{NATA}$ did not change with ATP-Mg$^{2+}$ while F increased, resulting in a relative fluorescence increase. Data were plotted as a function of ligand concentration (Fig. 1A, C for example).

For ATP-Mg$^{2+}$ binding in presence of R6G, Doxorubicin, or Hœchst33342, ligands were dissolved in water and added to a final concentration of 50 μM for R6G, 10 μM for Hoechst33342 and 300 μM for Doxorubicin, and the same experiment was carried out. For all ligands, the relative fluorescence-change due to ATP-Mg$^{2+}$ binding is an increase in fluorescence, as opposed to fluorescence quenching for ligand binding alone (see below).

## Ligand binding

R6G, Doxorubicin and Hœchst33342 binding were carried out by probing the intrinsic fluorescence change as a function of ligand concentration. Fluorescence was recorded on a SAFAS Xenius spectrophotofluorimeter set up at a constant photo multiplicator voltage of 570 V at 0.5 μM BmrA dimer, with the same settings and experiment as descried above. For Hoechst33342 that could absorb energy in the emission zone of tryptophan residues, no FRET was observed so the binding affinity was measured also using relative intrinsic tryptophan quenching.

## Reconstitution into liposomes

300 μL of *E. coli* total lipid extract (Avanti Polar lipids) in chloroform at 25 mg/mL were placed into a 50-mL glass balloon and evaporated using a gentle nitrogen stream while turning the balloon to create a monolayer, under a hood. The lipid film was resuspended and solubilized by addition of 300 μL of lipid buffer (20 mM HEPES-NaOH pH 7.5, 100 mM NaCl) supplemented by 75 μL DDM at 10 % (w/v), and vortexing at high speed for 5 min. The balloon was then placed on a gentle rocker at room temperature for 1 h. BmrA purified in the DDM/Cholate mixture was added to 100 μL of solubilized lipids to ensure a ratio 1 BmrA dimer per 3000 lipids (considering an average MW$_{lipids}$ = 750 g/mol, e.g., 40 μL of BmrA at 2.8 mg/mL). The solution

was completed to 500 μl with lipid buffer and incubated 45 min. at room temperature under gentle agitation. Detergents were then removed by 3 additions of 40 mg pre-activated biobeads, each followed by an incubation of 1 h at room temperature under gentle agitation between each biobead addition. The final proteoliposomes solution was then extruded by passing 11 times through a 400-nm membrane filter (Avanti Polar Lipids). These final proteoliposomes were kept at 4 °C and used immediately. BmrA retains its activity during 1 week in these conditions.

## ATP hydrolysis assay

The ATPase activity of BmrA was measured as previously described[12]. The protein in solution in 20 mM HEPES-NaOH pH 7.5, 100 mM NaCl, 0.7 mM DDM and 0.7 mM Na cholate was diluted in the ATPase activity assay buffer containing a mixture of 0.7 mM DDM and 0.7 mM Na cholate, and the ATPase activity recorded. ATP hydrolysis was measured using an enzymatic coupled assay, where ADP is regenerated in ATP by the Pyruvate kinase at the expense of PEP and generating pyruvate. Pyruvate is reduced into lactate by the Lactate dehydrogenase by oxidizing NADH into NAD + . The disappearance of NADH was followed by absorbance at 340 nm, indicative of ATP hydrolysis. Since BmrA has been shown to display an activation of ATPase activity[10], and that it can also be observed in the assays performed here and displayed in Fig. 4, initial (i.e., 20–30 s after ATP-Mg$^{2+}$ addition) and maximum (towards the end of the measurement) velocity were recorded. To represent the increase in velocity, the difference between velocities in presence and absence of R6G were normalized to the velocity in absence of R6G. ATPase activities were conducted at fixed ATP-Mg$^{2+}$ concentration and varying R6G concentrations, or at fixed R6G concentration and varying ATP-Mg$^{2+}$ concentrations. All ATP hydrolysis assays were realized with 4 μg wild-type and E504A inactive mutant in different amphipathic environments (detergent, liposome, inverted membrane vesicles). R6G was pre-incubated with the protein at 50 μM for 10 min at 25 °C.

## Transport experiments

Transport assays followed the fluorescence of Hœchst 33342 (Sigma-Aldrich) or doxorubicin (Sigma-Aldrich) during their transport by BmrA across the membrane with different ATP-Mg$^{2+}$ (Sigma-Aldrich) concentrations. The set-up experiment consisted of 50 μg *E. coli* inverted membrane vesicles containing over-expressed BmrA mixed with 1 μM of fluorescent molecules (Hœchst 33342 or doxorubicin, stock solution resuspended in water). Transport was initiated by adding a certain concentration of ATP-Mg$^{2+}$ and monitored at 25 °C in 1 mL quartz cuvettes, recording the fluorescence on Xenius fluorimeter (SAFAS) at 468 nm with a bandwidth of 10 nm upon excitation at 355 nm with a bandwidth of 10 nm for Hoechst 33342, and at 593 nm upon excitation at 468 nm with the bandwith of 10 nm for excitation and emission. Different ATP-Mg$^{2+}$ concentrations were tested, 20, 50, 100, 200, 300, 700, 1000, and 2000 μM. The transport buffer was made of 50 mM HEPES-NaOH pH = 7.5 (Sigma-Aldrich), 8.5 mM NaCl (Sigma-Aldrich), 5 mM MgCl$_2$ (Sigma-Aldrich), supplemented by 5 mM NaN$_3$ (Sigma-Aldrich) and 2 mM Na$_2$S (Sigma-Aldrich) which are inhibitors of the electron transport chain. Transport assays with E504A inactive mutant were also performed to visualize some potential non-specific transport. All experiments were done in duplicate of duplicates. Standard curves were performed to estimate the amount of drug transported. They were performed on the same inverted membrane vesicles, or in pure liposomes with a fixed lipid concentration (The liposomes concentration was set to reach the same maximum fluorescence for the ligand as in membrane vesicles, which resulted in 10 mg/ml lipids). The curves were performed from zero to the maximum amount of substrate added, in an incremental manner.

## Normalization of transported substrate/ATP hydrolyzed

Both transport assays and ATPases measurement were only considered for the first 15 s following ATP-Mg$^{2+}$ addition, where the measurements are linear as seen in Supplementary Figs. 29B and 30B. For the transport assay, fluorescence for the WT was subtracted from the mutant equivalent at each ATP-Mg$^{2+}$ concentration. Using this normalized fluorescence, the quantity of ligand transported was deduced using the standard curve in Supplementary Figs. 29G and 30G, yielding nmol of transported substrate / s. For ATPase measurement, a separate experiment was conducted on the same sample. Here again, activity was deduced from the residual activity seen for the E504A mutant, yielding BmrA-sensitive activity in nmol ATP /s. For each ATP-Mg$^{2+}$ concentration, the value of transported substrate was divided by the ATP hydrolyzed to make the curves in in Supplementary Figs. 29H and 30H and Fig. 4F. Importantly for reproducibility matters, all the measurements for the whole range of ATP-Mg$^{2+}$ concentration (one monoplicate) were conducted the same day on the same sample (in Supplementary Figs. 29C–G and 30C-G).

## Reporting summary

Further information on research design is available in the Nature Portfolio Reporting Summary linked to this article.

## Data availability

The structure coordinates have been deposited in the Protein Data Bank and EMDB under the following accession codes; E504A$^{apo}$ PDB :8REZ, EMD-19113. E504A$^{apo-25μMATP}$ IF: PDB :8RGA], EMD-19131; OF: PDB: 8RIA, EMD-19131. E504A$^{apo-100μMATP}$ PDB :8RI1, EMD-19180. E504A$^{R6G}$ PDB :8RF1, EMD-19115. E504A$^{R6G-25μMATP}$ PDB :8RG7, EMD-19130. E504A$^{R6G-70μMATP}$ PDB :8RGN, EMD-19135. E504A$^{H33342}$ PDB :9GSJ, EMD-51550. The molecular dynamics files are available at: https://doi.org/10.5281/zenodo.14222107. PDB codes of previously published structures used in this study are 6R72, 6R82 and 6R81. Source data are provided with this paper.

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

## Acknowledgements

We acknowledge the European Synchrotron Radiation Facility for provision of beam time on CM01 and we would like to thank all the staff for assistance. We thank FRISBI (PID-169 and -202) and INSTRUCT (PID-24618) for funding access to microscopes. This work used the platforms of the Grenoble Instruct-ERIC center (ISBG; UAR3518 CNRS-CEA- UGA-EMBL) within the Grenoble Partnership for Structural Biology (PSB), supported by FRISBI (ANR- 10-INSB-05-02 & project ID 160 to P.B.) and GRAL, financed within the University Grenoble Alpes graduate school (Écoles Universitaires de Recherche) CBH-EUR-GS (ANR-17-EURE-0003). The electron microscope facility is supported by the Auvergne-Rhône-Alpes Region, the Fondation pour la Recherche Médicale (FRM), the fonds FEDER and the GIS-Infrastructures en Biologie Santé et Agronomie (IBiSA). We thank Daouda Traoré for personal time on the ESRF CM01 Titan Krios. This project was funded by ANR-19-CE11-0023-01 for C.O., J.M.J., V.C., P.F., A.G., and S.M., and ANR-23-CE11-0031-01 for V.C., P.F., S.M., L.M., L.Z., and G.S. The authors wish to thank Alexis Michon with his constant IT help during the course of this article. We gratefully acknowledge support from the CNRS/IN2P3 Computing Center (Lyon - France) for providing computing and data-processing resources needed for this work.

## Author contributions

V.C. initiated the study. A.G. and L.M. expressed all the proteins, purified them and prepared them for cryoEM data acquisition and enzymologic characterization. A.G. processed all the data for the apo form and R6G-bound proteins, constructed all the models and performed 3DVA calculations and movies. A.G. performed the ATP binding assay for E504$^{apo}$ and in presence of R6G. L.M. performed all the other biochemical and enzymologic analysis. S.M. prepared protein and membranes. E.Z. and G.S. prepared all the grids and observed them on the IBS GLACIOS with overnight data collection for 2D classes, and performed some data collection on the CM01 KRIOS. C.O. and J.M.J. performed early analysis on ATPase stimulation by drugs at low ATP concentrations. C.O. and J.M.J. shared the WT and A582C structures. E.B. and R.T. performed MD simulations and logP analysis. J.M. performed analysis of the varref bundles. V.C. performed the variability refinement analysis. P.F. and V.C. supervised the whole project. All authors participated in writing the manuscript.

## Competing interests

The authors declare no competing interests.
