## [Transparent Peer Review file · Nature Communications]

Rhodamine6G and Hoechst33342 narrow BmrA conformational spectrum for a more efficient use of ATP

Corresponding Author: Dr Vincent Chaptal

Version 0:

Reviewer comments:

Reviewer #1

(Remarks to the Author)

Manuscript by Gobet et Moissonnier et al. reports on a joint experimental and computational investigation of the proposed allostery between substrate and ATP-Mg²⁺ binding, triggering the IF-to-OF transition of a bacterial ABC transporter, namely BmrA. By considering NBD dynamics through a powerful analysis of cryo-EM density maps and supported by enzymology and MD simulations, they highlight the allosteric communication between the substrate binding site and nucleotide binding sites (NBS). Their findings strongly suggest that substrate binding might favor ATP binding in an asymmetric fashion. In other words, they suggest that substrate binding helps facilitate the initial ATP binding, which, in turn, supports the second ATP binding required for the IF-to-OF transition. The present study involves an exhaustive and reliable amount of work encompassing cryo-EM resolution and experiments, enzymology, and MD simulations, leading to significant advances in understanding ABC function. More importantly, their findings are clearly relevant for deciphering the relationship between milestone steps and their order in the ABC transport cycle sequence. However, the present study could benefit from more detailed analyses and discussion (see comments below), as well as more details regarding MD simulations that appear underused.

i) The "Material & Methods" section regarding MD simulations is unclear and appears "unfinished". The process of protein insertion into the membrane is not adequately explained. Was it achieved using the GROMACS package (if yes, which software) or Packmol-Memgen? For the sake of transparency and reproducibility, more details are required regarding minimization, thermalization, and box equilibration prior to MD productions. The authors claim the use of NPT, which is likely not the case. Indeed, simulations carried out in a membrane must be conducted under at least semi-isotropic conditions. If this is not the case, the MD simulations may not be suitable and trustworthy, and they must be re-performed under suitable conditions. Likewise, which thermostat was used for MD simulation? Finally, more details are required for the force field since "Amber36" does not exist (as far as I know). Which force field was used to model lipids (Slipids, Lipid17, Lipid21)? Likewise, "standard Amber 14:SB" (which might mean FF14SB) is mentioned as well as "Amber14:EHT". This is confusing. Which force field was used for the ATP molecule? Was it based on GAFF2 or derived from DNA.OL15? Likewise, how was R6G parameterized? Adding ATP and R6G topologies and parameters to the Electronic Supplementary Information (ESI), as well as initial and final structures, could be important for the community.

ii) MD simulations were conducted for 700 ns. However, it is not stated in the manuscript which part of the production runs the analyses were conducted on. Figure 3D suggests that the "protein" equilibration along production runs took approximately 125 frames (i.e., 250 ns). Therefore, analyses should be conducted only on the last 450 ns. It would be also fruitful for the community to provide more details about how MD analyses were performed, explicitly mentioning tools that were used and parameters.

iii) Recent studies have investigated allosteric communications in membrane proteins from MD simulations using different relevant approaches (Allopath tools or network analyses, see e.g., 10.1063/5.0020974, 10.1039/d0sc06288j, 10.1038/s42003-023-04537-3, and others). Further investigations considering such approaches or at least dynamic cross-correlation along proteins might provide relevant insights supporting experimental observations by proposing key regions playing a role in allostery.

iv) Given the very interesting asymmetric allosteric behavior suggested in the present study, the authors may also discuss an evolutionary perspective, especially regarding NBD-degenerated ABC transporters.

Minor comments:

- P3 "transistion" should be "transition".
- For many supplementary and main draft figures, axis labels are barely readable (see e.g., Figure S13C, S17, etc...).
- 2D-RMSD (S17) is interesting; however, per-residue RMSF could also provide interesting information to correlate with cryo-EM-based dynamics.
- Figure S20 reports residue displacement during MD simulation. Again, RMSF may be more adapted to depict the flexibility. Furthermore, which reference was used as the "initial" structure for measuring the displacement?
- Figure S19 represents the distance between the initial and final positions, but what if it moves in several directions over the simulations? Did the authors consider PCA to extract the main sources of structural variability?
- The sentence "Each frame is sampled each 2ns, for a total of 700ns for each simulation" is unclear. Can the authors clarify it? Were snapshots extracted every 2 ns from simulations lasting 700 ns?
- It is important that a table summarizes all information regarding the system size and components (number of lipids, size of the initial and post-equilibration/production boxes, number of water per lipids, etc.).
- The authors state that simulations lasted 700 ns per replica, and snapshots were saved every 200 ns. It should be 2 ns to finally get 350 frames.

Reviewer #2

(Remarks to the Author)

General comments

The study by Gobet et al presents a series of cryo-EM structures of the bacterial ABC exporter BmrA, obtained at different concentrations of ATP, either in the absence or in the presence of the transport substrate Rhodamine6G (R6G). The structural data are complemented with ATP binding, ATPase, and transport assays. The authors find that the presence of R6G renders ATP binding to the two ATP binding sites positively cooperative, and reduces structural flexibility of the transporter. As I am not qualified to assess the structural aspects of the study, my comments are focused exclusively on the functional aspects and the overall presentation.

In its present form the paper is not written for a broad readership. The background and the hypotheses being tested are not clearly explained, and it does not become clear to the reader what are truly novel conceptual advances. For the same reason, the validity of several of the authors' claims is difficult to assess: the offered explanations – within the context of the present study – fall short of supporting them. That does not necessarily mean that these statements are incorrect, but more information would be required to convince the non-expert reader. Furthermore, insufficient methodological detail is provided to allow assessment of the validity of the quantitative conclusions. All in all, I find it difficult to judge whether the scientific advance is substantial enough to warrant publication in Nature Communications.

Specific comments

- Line 106: "addition of the substrate R6G... resulted in a shift towards a sigmoidal-type curve". What is the Hill coefficient?
- Line 108P: "meaning that... the NBDs do not bind ATP-Mg²⁺ the same way anymore"
This argument is unclear. Positive cooperativity in itself does not necessarily imply asymmetry. The best counter-example is the Monod-Wyman Changeux model, also called the "symmetry model", which the authors themselves refer to (lines 360-366). In the MWC model each subunit may exist in two states (T and R), and substrate binding to a subunit shifts the conformational equilibrium from T towards R, but the T-to-R transition happens in a concerted manner in all subunits, preserving the overall symmetry of the oligomer at all times. Thus, asymmetry (also mentioned in the abstract, line 36) does not follow from this result.
- Line 109: "the pre-binding of R6G increased the apparent affinity for ATP, together with the steep transition suggesting that R6G binding ensures a more efficient conversion to the OF conformation mediated by ATP-Mg²⁺ binding."
This argument is also unclear. The sigmoidicity of the ATP binding curve implies positive cooperativity between the two ATP binding events. But why this would imply more efficient conversion to the OF conformation remains unexplained.
- Fig. 1A and C. What is shown on the y-axes of these graphs? The Methods section does not provide any information on the ATP binding assay.
- Line 123: "This is in good agreement with the allostery model"
Please describe what exactly is meant by the allostery model for BmrA. Also, if the authors' aim is to contrast/compare this particular model with other existing models, please explain the differences between those models and their predictions, so that the reader can evaluate to what extent the presented data are in favor of one particular model.
- Figure 4F. The authors find that the stoichiometry of transported substrate per ATP hydrolyzed depends on ATP concentration in a biphasic manner, and use this finding to argue that transport and ATP hydrolysis are uncoupled. But how were the transport assays quantified? In particular, how was the transport rate normalized to the rate of ATP hydrolysis? These are very important methodological details which are not described.
- Line 285: "the [E504A] mutant only undergoes the forward direction and is blocked in the OF conformation (Figure 1)"
This is certainly not true. If for the E504A mutant the OF state was indeed a "sink state", then at a protein:ATP molar ratio of

1:4 a uniform OF population would be observed. In contrast, a 52%-48% mixture of IF and OF is observed under such conditions (Fig. 1B).

8. ATP hydrolysis assay, line 542: "Initial and maximum velocity were recorded".

It is unclear what is meant by "initial" and "maximum" velocity. Since the ATP regenerating system keeps [ATP] fixed, the velocity is expected to remain constant, as is also apparent on the curves shown in Fig. 4B-C. Do the authors mean velocity in the absence vs. presence of R6G?

Reviewer #3

(Remarks to the Author)

The manuscript „R6G narrows BmrA conformational spectrum for a more efficient use of ATP”, submitted by Gobet et al., suggests a mechanism through which Rhodamine6G (R6G) binding leads to an allosteric ATP binding mode of the ABC transporter BmrA.

The authors apply multi-model cryoEM, MD simulation, and activity assays to confirm a conformational change in the hydrolysis deficient BmrA mutant E504A in the presence and absence of R6G by cryo-electron microscopy and MD simulation, resulting in the appearance of an outward-facing conformation at lower ATP-Mg²⁺ concentrations in the presence of R6G. Furthermore, ATP binding at increasing ATP-Mg²⁺ reveals an increased binding affinity for the E504A mutant at low ATP-Mg²⁺ concentrations.

The results are detailed and structurally illustrate the correlation of ATP binding at low ATP-Mg²⁺ concentration to the conformational change of BmrA upon R6G binding. However, several strong concerns and issues should be addressed and clarified before publication is possible. Furthermore, the text would benefit greatly from rewording/formatting and, especially, shortening of sentences. Also, some sections appear unnecessarily long (such as 2.3).

Overall, the manuscript is a bit disappointing as there clearly is a lot of potential, but too many things distract from the story or are incomprehensible. Of note, in an earlier manuscript the authors (Ref 13) already presented a lot of the findings described here.

1. I have a major problem with the conclusions drawn from the data presented in Figure 1 – to my understanding, this is the key figure of the manuscript. To me, this figure should illustrate that due to the addition of R6G, the chances of NBD dimerization of the transporter are increased at lower ATP concentrations, and it illustrates the allosteric binding.

According to graphs A and C, the amount of ATP binding is almost identical at 25 μM – in fact, the unstimulated version is higher. The authors state that “the pre-binding of R6G increased the apparent affinity for ATP, together with the steep transition suggesting that R6G binding ensures a more efficient conversion to the OF conformation mediated by ATP-Mg²⁺ binding”. For the second point (25 μM), I do not understand this.

Moreover, I have a general problem with this interpretation. The authors state: “In contrast, in the absence of R6G (E504Aapo-25 μMATP), only IF reconstructions could be observed as ATP-Mg²⁺ will distribute equally among both NBDs and will not yield enough particles in the OF conformation to be seen (Figure 1B).”

For ATP-binding, one would expect 25% unbound (0/0), 25% and 25% single bound (0/1 or 1/0), and 25% double bound NBDs (1/1), the latter (25%) should result in an OF conformation. As the authors state: no OF was detected in the unstimulated version. However, with 1,7 mio particles, even a smaller ratio than 25% should be detectable (10% should work). And even with less stimulation (R6G second circle) 25% OF conformations are detected. All of this seems counterintuitive. Also the ratio for the third circle is 60/40 vs 48/52, but here, the stimulated circle is much higher than the unstimulated one – why is the outcome and correlation between EM and biochemistry here different?

In any case, the achieved resolution of just below 4 Å is rather low given the large number of particles. It is possible to obtain a higher resolution (3.3), as shown by the authors in their apo experiment. This indicates that the volume is still a combination of many different conformations – presumably also OF.

It would be extremely helpful for their claims if the authors could investigate ATP binding to IF conformations. At 3.3 Å, it should certainly be possible to identify nucleotides at the binding pockets and even to sort for relative presence and absence. As hydrolysis will not really occur it is not important to discriminate between ATP and ADP.

2. The authors present cryo-EM structures of E504Aapo and E504AR6G in detergent and MD simulations in a lipid bilayer but do not mention if there is a possibility that the detergent micelle or the absence of lipids influence the conformational change in the cryo-EM density map. Also, differences during the change of the hydrophobic environment could be seen in the ATPase activity results, showing differences between the detergent and liposome/membrane graph in Figure 4D.

3. The concentration of the substrates added in the cryo-EM sample and the activity/transport assay are not comparable, and since the substrates Hoechst33342 and Doxorubicin were added for comparison, are conformational changes in the same TM helices expected as for R6G?

4. It would have been optimal if the same substrate used for the structural analysis had been tested for transport properties. R. Ernst et al., 2008 (DOI 10.1016/j.jmb.2008.05.074) present R6G transport in yeast membrane. Is it possible to adapt this for membrane vesicles?

5. In the introduction, the authors write: “allowing cells to further adapt by acquiring target mutations” – Does this indicate

active evolution?

6. Figure 3 is very cluttered. Would it not be much better to only show the 2 extremes and an average or something? Also, I am not sure what I am looking at; the black lines are almost invisible against the background ... Furthermore, it is very bold to claim a 1 Å shift at this rather moderate resolution. This is clearly an overinterpretation of the available data.

7. I am not convinced that the data presented supports the finding claimed in the last sentence of the abstract: that the diffusion rate is the rate limiting step of the reaction.

8. In their introduction the authors establish the apo IF conformation as resting state for all type IV ABC transporters. I am not at all convinced that this is true or has been shown – I am not even sure this has been entirely shown for one of these transporters. Later in the introduction the authors themselves relate their statement writing that the transporter swings back and forth around an occluded state. Somehow these sentences do not add up.

Minor changes are listed below. Especially the Materials & Methods section contains numerous spelling and grammar errors that must be corrected.

- An inadequately chosen colour scheme compromises the clarity of Figure 2B.
- Line 126: what does the insertion of ..., being 50% of the ATP-Mg₂₊ in the sample, ... mean, or why is this important information?
- Line 161: missing space
- Line 166: the wording of ... does not occur within a monomer is misleading since it occurs in only one monomer
- Line 177: With cryoEM structure resolution, ...
- Line 206: which TM helices are kinked
- Line 207-208: ... of the NBD and as a continuation of the TM helices...
- Line 229: what is the ATP-Mg₂₊ concentration
- Line 231: ... in a different range ...
- Figure 4D: preparation of BmrA in the “membrane” environment is missing
- Caption Figure 4F: a range of ATP-Mg₂₊ was sampled in D/, not C/
- Figure 5A: it should be 2x Pi released, and the arrow visualizing substrate release indicates a substrate release without ATP hydrolysis, which is in conflict with the performed cryo-EM experiments using E504A
- Caption Figure 5A: it is a yellow/orange circle or oval, not a cylinder
- Line 366: ... to understand the structure/function relationship ...
- Line 373: ... 6-histidine tag at the N-terminal.
- Line 374: ... at 37°C...
- Line 377/378/381: The culture ... / The pellet... / The supernatant ...
- Line 382/384: pH 8.0, missing space
- Line 387: ... incubated for 1 h and 20 min ...
- Line 390: ..., and 20 mM imidazole.
- The molar ratio of ATP and Mg₂₊ is not stated
- Line 461: ... the NBDs undergo ...
- Line 484-487: to be deleted. These sentences are repeated in lines 488-491
- Line 506: Vincristine binding was not presented or mentioned in the results, but in Supp-Figure 25 and therefore, Vincristine binding is not part of the story
- Line 515: which figure shows the mentioned data as a function of ligand concentration, or ATP as the ligand has to be defined
- Line 520: the phrase monolayer is not used correctly; a lipid film is created
- Line 525: ... to ensure a ratio of 1 BmrA dimer per 3000 lipids ...
- Line 555: ... with 1 μM ...
- Line 559-561: repetitions of the phrase bandwidth of 10nm
- Line 561-562: wrong grammar
- Line 572: ..., in small increments ..., small is not defined
- Caption Supp-Figure 13A: sentence is unstructured

Reviewer #4

(Remarks to the Author)

Reviewer #5

(Remarks to the Author)

Version 1:

Reviewer comments:

Reviewer #1

(Remarks to the Author)

Overall, the manuscript has improved, and the authors have addressed most of the comments. However, the molecular dynamics (MD) section, including how it was performed, remains unclear to me:

Point ii)

In their rebuttal letter, and upon reviewing Figure 4B, the authors acknowledge that equilibration (from the so-called “production run”) took 250 ns. Therefore, the analyses should be conducted from $t = 250$ ns onwards. I would like to remind the authors that there should be no confusion between thermalization/box equilibration and system equilibration—the latter is system-dependent and should not be used as the basis for conformational analyses. Additionally, the x-axis label in Panel 4D should represent MD simulation time in nanoseconds (ns), not “Frame.”

Point iii)

I am disappointed that the authors did not explore allosteric mechanisms through MD simulations, especially since the experimental data presented in this manuscript suggests potential pathways that could be supported by such simulations. However, I understand that this might have required more effort than the authors were able to allocate. Additionally, Reference 37, which was added in response to the review, is not appropriate for ABC transporters.

Reviewer #2

(Remarks to the Author)

The authors have addressed all my concerns. I have no further comments.

Reviewer #3

(Remarks to the Author)

The authors have appropriately addressed the raised concerns. I have no further comments.

Reviewer #4

(Remarks to the Author)

Reviewer #5

(Remarks to the Author)

Manuscript number: NCOMMS-24-20516-T

Dear Reviewers,

We would like to thank and acknowledge your work on our manuscript.

We carefully took into account all comments, which led us to substantially modify the manuscript. We notably added new cryoEM data of BmrA in complex with a new substrate (Hoechst33342) and performed variability analysis. Importantly, this new data shows a similar pattern of conformational space reduction for the NBDs, albeit with some substrate specificity, thereby strengthening the findings presented in this manuscript. Following this addition, we significantly modified the text, we divided the cryoEM variability and Molecular Dynamics simulations and added 4 new supplemental figures and 1 table. We also clarified the allosteric role of substrate on the NBDs, which lead to a cooperative binding for ATP, and distinguish these two effects in the text.

We hope that these modifications will convince you of the quality of our findings.

Detailed point-to-point answers are appended below.

Thank you.

Best regards

The co-authors.

REVIEWER COMMENTS

Reviewer #1 (Remarks to the Author):

Manuscript by Gobet et Moissonnier et al. reports on a joint experimental and computational investigation of the proposed allostery between substrate and ATP-Mg²⁺ binding, triggering the IF-to-OF transition of a bacterial ABC transporter, namely BmrA. By considering NBD dynamics through a powerful analysis of cryo-EM density maps and supported by enzymology and MD simulations, they highlight the allosteric communication between the substrate binding site and nucleotide binding sites (NBS). Their findings strongly suggest that substrate binding might favor ATP binding in an asymmetric fashion. In other words, they suggest that substrate binding helps facilitate the initial ATP binding, which, in turn, supports the second ATP binding required for the IF-to-OF transition. The present study involves an exhaustive and reliable amount of work encompassing cryo-EM resolution and experiments, enzymology, and MD simulations, leading to significant advances in understanding ABC function. More importantly, their findings are clearly relevant for deciphering the relationship between milestone steps and their order in the ABC transport cycle sequence. However, the present study could benefit from more detailed analyses and discussion (see comments below), as well as more details regarding MD simulations that appear underused.

i) The "Material & Methods" section regarding MD simulations is unclear and appears "unfinished". The process of protein insertion into the membrane is not adequately explained. Was it achieved using the GROMACS package (if yes, which software) or Packmol-

Memgen? For the sake of transparency and reproducibility, more details are required regarding minimization, thermalization, and box equilibration prior to MD productions. The authors claim the use of NPT, which is likely not the case. Indeed, simulations carried out in a membrane must be conducted under at least semi-isotropic conditions. If this is not the case, the MD simulations may not be suitable and trustworthy, and they must be re-performed under suitable conditions. Likewise, which thermostat was used for MD simulation? Finally, more details are required for the force field since "Amber36" does not exist (as far as I know). Which force field was used to model lipids (Slipids, Lipid17, Lipid21)? Likewise, "standard Amber 14:SB" (which might mean FF14SB) is mentioned as well as "Amber14:EHT". This is confusing. Which force field was used for the ATP molecule? Was it based on GAFF2 or derived from DNA.OL15? Likewise, how was R6G parameterized? Adding ATP and R6G topologies and parameters to the Electronic Supplementary Information (ESI), as well as initial and final structures, could be important for the community.

We would like to thank the reviewer for catching this and apologize for this mix-up between versions. We have substantially modified and updated this section of the new M&M, which should answer all of the reviewer's rightful points.

ii) MD simulations were conducted for 700 ns. However, it is not stated in the manuscript which part of the production runs the analyses were conducted on. Figure 3D suggests that the "protein" equilibration along production runs took approximately 125 frames (i.e., 250 ns). Therefore, analyses should be conducted only on the last 450 ns. It would be also fruitful for the community to provide more details about how MD analyses were performed, explicitly mentioning tools that were used and parameters.

The equilibration indeed took 250ns but was not shown. The data presented is only for the production runs and were thus analyzed from the start. We have modified the text, M&M and figure legend to avoid ambiguity.

iii) Recent studies have investigated allosteric communications in membrane proteins from MD simulations using different relevant approaches (Allopath tools or network analyses, see e.g., 10.1063/5.0020974, 10.1039/d0sc06288j, 10.1038/s42003-023-04537-3, and others). Further investigations considering such approaches or at least dynamic cross-correlation along proteins might provide relevant insights supporting experimental observations by proposing key regions playing a role in allostery.

Indeed, the literature is rich on ABC transporters and their allostery. These papers investigate the role of ligands on ABCB1 or MRP4 deformations, as well as the role of external lipids on membrane protein deformation. While these papers are solely investigating allostery by means of MD simulations, we couldn't reproduce the same type of investigation here as it would result in too much information and take away from the main message we are trying to convey. It should be a whole study in itself, while MD simulations were more used here in backing up the experimental cryoEM data. Nevertheless, we added the references in the discussion to point the reader to these studies.

iv) Given the very interesting asymmetric allosteric behavior suggested in the present study,

the authors may also discuss an evolutionary perspective, especially regarding NBD-degenerated ABC transporters.

We agree with the reviewer that our study hints towards ABC transporters with degenerated NBDs. However, we felt like adding this discussion to the manuscript would divert too much from the main message, which is also what reviewer #2 and #3 point to, so we rather not include it on the discussion. Thank you for point it out though.

Minor comments:

- P3 "transistion" should be "transition". Thank you, it has been corrected.
- For many supplementary and main draft figures, axis labels are barely readable (see e.g., Figure S13C, S17, etc...). We have expanded the axis on Figure S17.
- 2D-RMSD (S17) is interesting; however, per-residue RMSF could also provide interesting information to correlate with cryo-EM-based dynamics.

We have added Supplemental-Figure 20 to show the fluctuations of both proteins during the simulation and added a reference in the new section 2.4 on MD simulations.

- Figure S20 reports residue displacement during MD simulation. Again, RMSF may be more adapted to depict the flexibility. Furthermore, which reference was used as the "initial" structure for measuring the displacement?

For Figure S20 (now S19) the reference was the initial frame of the simulation. The figure legend was modified to include it.

- Figure S19 represents the distance between the initial and final positions, but what if it moves in several directions over the simulations? Did the authors consider PCA to extract the main sources of structural variability? Indeed, this representation is a very simplistic view of the overall displacements (which are otherwise shown several times in the manuscript, and in Figure 4). As suggested, we have added a PCA analysis (Figure S21) of the movements, and the section in a M&M.

- The sentence "Each frame is sampled each 2ns, for a total of 700ns for each simulation" is unclear. Can the authors clarify it? Were snapshots extracted every 2 ns from simulations lasting 700 ns? Yes, this is what we meant. We have modified it to be more explicit.

- It is important that a table summarizes all information regarding the system size and components (number of lipids, size of the initial and post-equilibration/production boxes, number of water per lipids, etc.). The table has been included (Supp-Table 2).

- The authors state that simulations lasted 700 ns per replica, and snapshots were saved every 200 ns. It should be 2 ns to finally get 350 frames. This was a typo and has been corrected in the M&M section and the figure S19 legend.

Reviewer #2 (Remarks to the Author):

General comments

The study by Gobet et al presents a series of cryo-EM structures of the bacterial ABC exporter BmrA, obtained at different concentrations of ATP, either in the absence or in the presence of the transport substrate Rhodamine6G (R6G). The structural data are complemented with ATP binding, ATPase, and transport assays. The authors find that the presence of R6G renders ATP binding to the two ATP binding sites positively cooperative, and reduces structural flexibility of the transporter. As I am not qualified to assess the structural aspects of the study, my comments are focused exclusively on the functional aspects and the overall presentation.

In its present form the paper is not written for a broad readership. The background and the hypotheses being tested are not clearly explained, and it does not become clear to the reader what are truly novel conceptual advances. For the same reason, the validity of several of the authors' claims is difficult to assess: the offered explanations – within the context of the present study – fall short of supporting them. That does not necessarily mean that these statements are incorrect, but more information would be required to convince the non-expert reader. Furthermore, insufficient methodological detail is provided to allow assessment of the validity of the quantitative conclusions. All in all, I find it difficult to judge whether the scientific advance is substantial enough to warrant publication in Nature Communications.

We thank the reviewer for his comments. We have substantially modified the manuscript and hope that it will convince the reviewer of the quality and novelty of these findings. More specifically:

- The background and the hypotheses being tested are not clearly explained:
 - improvement: We modified the introduction to explain the rationale of the study.
- it does not become clear to the reader what are truly novel conceptual advances
 - improvement: We added this phrase to the introduction: Altogether, this study reveals how protein flexibility visualized by cryoEM data can explain the enzymologic behavior of BmrA.
- the validity of several of the authors' claims is difficult to assess: the offered explanations – within the context of the present study – fall short of supporting them
 - improvement: The manuscript has been substantially modified and improved thanks to all reviewers' comments.
- more information would be required to convince the non-expert reader.
 - improvement: with the addition of new data and the modification of the manuscript, the conclusions are reinforced, which should help convince non-expert readers.
- insufficient methodological detail is provided to allow assessment of the validity of the quantitative conclusions
 - improvement: The M&M section has been substantially modified and detailed.

Specific comments

1. Line 106: "addition of the substrate R6G... resulted in a shift towards a sigmoidal-type curve".

What is the Hill coefficient?

The Hill coefficient is 3.4, it was stated in the text 2 lines below ($K_{0.5} = 70.0 \mu\text{M} \pm 2.6$, $n=3.4 \pm 0.3$), as well as in Figure S25 (now Figure S24) with the affinity measurements of several ligands. To avoid ambiguity, it is now replaced by "h". ($K_{0.5} = 70.0 \mu\text{M} \pm 2.6$, $h=3.4 \pm 0.3$)

2. Line 108P: "meaning that... the NBDs do not bind ATP-Mg²⁺ the same way anymore"
This argument is unclear. Positive cooperativity in itself does not necessarily imply asymmetry. The best counter-example is the Monod-Wyman Changeux model, also called the "symmetry model", which the authors themselves refer to (lines 360-366). In the MWC model each subunit may exist in two states (T and R), and substrate binding to a subunit shifts the conformational equilibrium from T towards R, but the T-to-R transition happens in a concerted manner in all subunits, preserving the overall symmetry of the oligomer at all times. Thus, asymmetry (also mentioned in the abstract, line 36) does not follow from this result.

Symmetry conversion in the MWC model is still a matter of debate and entering the debate is not the point of this manuscript. Thus, we removed the reference to this model in the discussion to avoid confusion, and focus more on the cooperativity of the NBDs for ATP, triggered by ligands. We thus simplified the message. We apologize for the confusion and we thank the reviewers #2 and #3 to help us streamline our message.

3. Line 109: "the pre-binding of R6G increased the apparent affinity for ATP, together with the steep transition suggesting that R6G binding ensures a more efficient conversion to the OF conformation mediated by ATP-Mg²⁺ binding."

This argument is also unclear. The sigmoidicity of the ATP binding curve implies positive cooperativity between the two ATP binding events. But why this would imply more efficient conversion to the OF conformation remains unexplained.

The cryoEM data show that the most important change remains the conformational change as the structures are 100% IF in initial apo conditions, and 100% OF when ATP is bound. The percentage of OF conformation is higher in presence of drugs compared to apo BmrA for lower ATP concentrations. We modified the text to make it clearer.

4. Fig. 1A and C. What is shown on the y-axes of these graphs? The Methods section does not provide any information on the ATP binding assay.

Sincere apologies, we indeed didn't include ATP binding experiments in the M&M, only ligand-binding. A new section of the methods has been added, as well as Figure 1 being modified to include "Relative Fluorescence increase".

5. Line 123: "This is in good agreement with the allosteric model"

Please describe what exactly is meant by the allosteric model for BmrA. Also, if the authors' aim is to contrast/compare this particular model with other existing models, please explain the differences between those models and their predictions, so that the reader can evaluate to what extent the presented data are in favor of one particular model.

As mentioned above, we distinguished more clearly the allostery of the drug on the NBDs, and the cooperativity observed for ATP binding.

To clarify, we have modified this phrase to: This is in good agreement with the cooperativity observed for ATP-Mg²⁺ binding, and the notion that the binding to one nucleotide-Binding Site (NBS) increases the affinity of ATP-Mg²⁺ for the second NBS (Figure 1D).

6. Figure 4F. The authors find that the stoichiometry of transported substrate per ATP hydrolyzed depends on ATP concentration in a biphasic manner, and use this finding to argue that transport and ATP hydrolysis are uncoupled. But how were the transport assays quantified? In particular, how was the transport rate normalized to the rate of ATP hydrolysis? These are very important methodological details which are not described.

We previously referred the reader to Supp-Figures 23 and 24 (now Supp-Figures 29 and 30) in the legend of Figure 4F (Now Figure 5F), together with a description of the methods of ATPase measurements and transport assays. Apparently, this was not clear enough so we have included a new section in the M&M (4.12) about how we carried out this experiment. We hope this clarifies the reviewer's question on this important experiment.

7. Line 285: "the [E504A] mutant only undergoes the forward direction and is blocked in the OF conformation (Figure 1)"

This is certainly not true. If for the E504A mutant the OF state was indeed a "sink state", then at a protein:ATP molar ratio of 1:4 a uniform OF population would be observed. In contrast, a 52%-48% mixture of IF and OF is observed under such conditions (Fig. 1B).

We thank the reviewer for this remark. The fact that not all the population shifts to OF in the 1:4 molar ratio is that the ATP concentration drops below the K_d and that ATP is not binding anymore. It also shows why there is an advantage of R6G binding as there is a higher percentage of OF conformation for a lower ATP concentration.

8. ATP hydrolysis assay, line 542: "Initial and maximum velocity were recorded".

It is unclear what is meant by "initial" and "maximum" velocity. Since the ATP regenerating system keeps [ATP] fixed, the velocity is expected to remain constant, as is also apparent on the curves shown in Fig. 4B-C. Do the authors mean velocity in the absence vs. presence of R6G?

For BmrA, we have previously shown an activation of the ATPase activity (Steinfeils et al, Biochemistry 2004, Figure 8A). This activation can also be observed on Figure 4B as the curves are not linear but show an acceleration as time goes; This acceleration is more clearly observed at 300 μM ATP-Mg²⁺ and much less for higher ATP concentrations. We then looked at the impact on initial (i.e. first 20-30 seconds after ATP addition) or maximum (i.e. towards the end of the measurement) velocities. We have modified this section to make it clearer in the manuscript.

Reviewer #3 (Remarks to the Author):

The manuscript „R6G narrows BmrA conformational spectrum for a more efficient use of ATP”, submitted by Gobet et al., suggests a mechanism through which Rhodamine6G (R6G) binding leads to an allosteric ATP binding mode of the ABC transporter BmrA. The authors apply multi-model cryoEM, MD simulation, and activity assays to confirm a conformational change in the hydrolysis deficient BmrA mutant E504A in the presence and absence of R6G by cryo-electron microscopy and MD simulation, resulting in the appearance of an outward-facing conformation at lower ATP-Mg²⁺ concentrations in the presence of R6G. Furthermore, ATP binding at increasing ATP-Mg²⁺ reveals an increased binding affinity for the E504A mutant at low ATP-Mg²⁺ concentrations.

The results are detailed and structurally illustrate the correlation of ATP binding at low ATP-Mg²⁺ concentration to the conformational change of BmrA upon R6G binding. However, several strong concerns and issues should be addressed and clarified before publication is possible. Furthermore, the text would benefit greatly from rewording/formatting and, especially, shortening of sentences. Also, some sections appear unnecessarily long (such as 2.3).

Thank you for your reviewing of the manuscript. With all the comments, we have substantially modified the manuscript, and “took the space” to explain more in detail some results. The section 2.3 has been divided in 2 to separate the cryoEM dynamics and Molecular dynamics.

Overall, the manuscript is a bit disappointing as there clearly is a lot of potential, but too many things distract from the story or are incomprehensible. Of note, in an earlier manuscript the authors (Ref 13) already presented a lot of the findings described here.

In Ref 13, we presented a static vision of the structures we are building on here, moving towards a dynamic vision based on experimental observations and reproduced in silico.

What is different and completely new in this study is that it presents a completely new way of observing the functioning of an efflux pump, revealing how the protein reacts and adapts structurally to the presence of the substrate to be transported.

1. I have a major problem with the conclusions drawn from the data presented in Figure 1 – to my understanding, this is the key figure of the manuscript. To me, this figure should illustrate that due to the addition of R6G, the chances of NBD dimerization of the transporter are increased at lower ATP concentrations, and it illustrates the allosteric binding.

This is exactly what the figure shows. At lower ATP concentrations, there are more OF in the presence of R6G. This is seen across the different points (ratio 1:1 and 70µM ATP).

According to graphs A and C, the amount of ATP binding is almost identical at 25µM – in fact, the unstimulated version is higher. The authors state that “the pre-binding of R6G increased the apparent affinity for ATP, together with the steep transition suggesting that R6G binding ensures a more efficient conversion to the OF conformation mediated by ATP-Mg²⁺”

binding”. For the second point (25uM), I do not understand this.

Moreover, I have a general problem with this interpretation. The authors state: “In contrast, in the absence of R6G (E504Aapo-25μMATP), only IF reconstructions could be observed as ATP-Mg²⁺ will distribute equally among both NBDs and will not yield enough particles in the OF conformation to be seen (Figure 1B).”

For ATP-binding, one would expect 25% unbound (0/0), 25% and 25% single bound (0/1 or 1/0), and 25% double bound NBDs (1/1), the latter (25%) should result in an OF conformation. As the authors state: no OF was detected in the unstimulated version. However, with 1,7 mio particles, even a smaller ratio than 25% should be detectable (10% should work). And even with less stimulation (R6G second circle) 25% OF conformations are detected. All of this seems counterintuitive. Also the ratio for the third circle is 60/40 vs 48/52, but here, the stimulated circle is much higher than the unstimulated one – why is the outcome and correlation between EM and biochemistry here different?

The reviewer raises here good points, on which we had many internal discussions including the exact points brought above. Let us respond point by point:

- For ATP-binding, one would expect 25% unbound (0/0), 25% and 25% single bound (0/1 or 1/0), and 25% double bound NBDs (1/1), the latter (25%) should result in an OF conformation.

Absolutely and this was our initial intention, and we hoped a “clever” way to obtain a single-bound ATP-Mg on the NBD and also we hoped to trap a transition state. However, it is not what is observed on the grid, although not far from it. Indeed, we have a high-resolution volume with the 2NBDs devoid of density for ATP, corresponding to the (0/0), and 3 volumes that are clearly IF. We have tried high and low to refine these volumes to higher resolution as we strongly believe that they represent the (0/1) and (1/0) single bound structures but couldn’t go further than the structures depicted in Figure 1B. It was equally frustrating to us as to the reviewer but we have to leave it at what it is. No OF conformations were seen, and the ab-initio volumes seen in supp-Figure 2 unambiguously show the absence of OF features. Even with this large number of particles from a very good grid, making that the absence of OF conformation is a solid observation.

The increase in intrinsic fluorescence caused by the addition of ATP-Mg results from changes in the environment of the protein's 6 tryptophan residues to varying degrees. The resulting mathematical modeling allows for determining the apparent affinity and binding mechanism. However, this method does not provide access to the corresponding conformational distribution, or at least only very approximately. CryoEM provides this information, although with a level of precision that only clearly distinguishes the two extreme IF and OF states, and not those where a single ATP-Mg is present. At 25 μM, even though fluorescence indicates an equivalent level of ATP-Mg binding with or without R6G, cryoEM unambiguously shows the conformational diversity of the two conditions.

We modified the text to incorporate this discussion.

- And even with less stimulation (R6G second circle) 25% OF conformations are detected. All of this seems counterintuitive. Also the ratio for the third circle is 60/40 vs 48/52, but here, the stimulated circle is much higher than the unstimulated one.

For the 60/40 (in presence of R6G), the point was gathered at lower ATP concentration than then one for the 48/52 (apo). This shows the effect of R6G in accelerating the transition and matches the curve very well.

In any case, the achieved resolution of just below 4 Å is rather low given the large number of particles. It is possible to obtain a higher resolution (3.3), as shown by the authors in their apo experiment. This indicates that the volume is still a combination of many different conformations – presumably also OF.

It would be extremely helpful for their claims if the authors could investigate ATP binding to IF conformations. At 3.3Å, it should certainly be possible to identify nucleotides at the binding pockets and even to sort for relative presence and absence. As hydrolysis will not really occur it is not important to discriminate between ATP and ADP.

The low resolution is a combination of multiple factors, as can be seen by the spread of resolution of all our datasets. The main factor in our opinion is the variability of the IF conformation of BmrA that we describe in section 2.3. For BmrA, we needed to have a large number of particles to go down in resolution, because of these many conformations. We tried of course to lower the number of particles, to perform focused refinements, to try different masks, etc... but we could never go below the resolution reported in each case. We wished to see the ATP as well but the data doesn't show it and shows features of the NBD (flexible loops) revealing the absence of bound nucleotide. As answered above and incorporated in the main text, the absence of OF conformation is a strong observation in itself, and OF particles do NOT hide inside the particle stack of the IF conformation.

Of note, even for the 3.3Å resolution apo, the NBDs are not the best resolved part as seen by the local resolution plots displayed in Supp-Figure 7.

2. The authors present cryo-EM structures of E504Aapo and E504AR6G in detergent and MD simulations in a lipid bilayer but do not mention if there is a possibility that the detergent micelle or the absence of lipids influence the conformational change in the cryo-EM density map. Also, differences during the change of the hydrophobic environment could be seen in the ATPase activity results, showing differences between the detergent and liposome/membrane graph in Figure 4D.

We added a part in the discussion to take this remark into account.

3. The concentration of the substrates added in the cryo-EM sample and the activity/transport assay are not comparable, and since the substrates Hoechst33342 and Doxorubicin were added for comparison, are conformational changes in the same TM helices expected as for R6G?

Following this query, we have substantially modified the manuscript.

We have collected the new dataset of E504 in complex with Hoechst33342 and performed 3D variability analysis. We have included 4 new supplemental figures, and restructured the whole text by splitting the “dynamics” section in 2, thus separating the MD simulation in section 2.4

and creating a new section 2.5 for this new Hoechst result. We modified the discussion accordingly.

The data are exactly in line with what was observed for R6G and strongly back up the claims already made. Thank you for suggesting this experiment.

4. It would have been optimal if the same substrate used for the structural analysis had been tested for transport properties. R. Ernst et al., 2008 (DOI 105 (13) 5069-5074) present R6G transport in yeast membrane. Is it possible to adapt this for membrane vesicles?

Indeed, it would have been nice to use R6G for transport experiments in membranes. Unfortunately, it is not possible despite many tries. In ref 13 where we already tried a lot, we performed whole cells assay to demonstrate R6G transport by BmrA, but we can't use this assay to quantify ATP consumption at the same time.

5. In the introduction, the authors write: "allowing cells to further adapt by acquiring target mutations" – Does this indicate active evolution?

To avoid confusion, we replaced the phrase by: Drug resistance mediated by ABC (ATP-Binding Cassette) transporters contributes to the first line of defense for organisms, decreasing the intracellular drug concentration below cytotoxic levels

6. Figure 3 is very cluttered. Would it not be much better to only show the 2 extremes and an average or something? Also, I am not sure what I am looking at; the black lines are almost invisible against the background ... Furthermore, it is very bold to claim a 1 Å shift at this rather moderate resolution. This is clearly an overinterpretation of the available data.

We agree that 1 Å displacement is very low and it should be taken with the whole range of displacements in mind. We can observe 2 different rotations and 1 translation. When R6G is bound, 1 rotation is decreased, and the translation is severely impaired. The 1 Å translation is in fact a fluctuation of the structure, a general vibration. Note that the addition of the new Hoechst data shows the same thing. We disagree that this is an overinterpretation of the data and we still stand by it.

7. I am not convinced that the data presented supports the finding claimed in the last sentence of the abstract: that the diffusion rate is the rate limiting step of the reaction.

We have removed this phrase from the abstract. We still keep a part in a discussion to speculate on this.

8. In their introduction the authors establish the apo IF conformation as resting state for all type IV ABC transporters. I am not at all convinced that this is true or has been shown – I am not even sure this has been entirely shown for one of these transporters. Later in the introduction the authors themselves relate their statement writing that the transporter swings back and forth around an occluded state. Somehow these sentences do not add up.

This was indeed a misleading phrase, we replaced it by: "MDR ABC transporters transport their substrates starting in an Inward-Facing (IF) conformation ..."

Minor changes are listed below. Especially the Materials & Methods section contains numerous spelling and grammar errors that must be corrected.

- An inadequately chosen colour scheme compromises the clarity of Figure 2B.

We tried to be consistent throughout the manuscript regarding colors, which can become very challenging with so much data. We have added the color in the figure legend for clarity.

- Line 126: what does the insertion of ..., being 50% of the ATP-Mg²⁺ in the sample, ... mean, or why is this important information?

We added a phrase to explain it as well as restructured the paragraph taking into account all of the reviewer's comments.

- Line 161: missing space

We have corrected it.

- Line 166: the wording of ... does not occur within a monomer is misleading since it occurs in only one monomer

We replaced it with “half-transporter”

- Line 177: With cryoEM structure resolution, ...

We have corrected it.

- Line 206: which TM helices are kinked

It is shown on Supp-Figure 14

- Line 207-208: ... of the NBD and as a continuation of the TM helices...

We simplified this phrase.

- Line 229: what is the ATP-Mg²⁺ concentration

Only 2 ATP-Mg²⁺ were added in the sites. The new MD simulation section 2.4 says it.

- Line 231: ... in a different range ...

We have corrected it.

- Figure 4D: preparation of BmrA in the “membrane” environment is missing

We would gladly modify it but we can't find what the referee is pointing to.

- Caption Figure 4F: a range of ATP-Mg²⁺ was sampled in D/, not C/

Thank you for catching this. We have corrected it.

- Figure 5A: it should be 2x Pi released, and the arrow visualizing substrate release indicates a substrate release without ATP hydrolysis, which is in conflict with the performed cryo-EM experiments using E504A

We have modified the 2 Pi to make it clearer. For substrate release, we believe that it occurs before ATP hydrolysis, as we discussed it in Ref 13 and our previous investigation of the OF conformation. This schematic of the transport mechanism is of course for the WT protein. We have added it in the figure legend.

- Caption Figure 5A: it is a yellow/orange circle or oval, not a cylinder

Thank you for catching this. We have corrected it.

- Line 366: ... to understand the structure/function relationship ...

We have corrected it.

- Line 373: ... 6-histidine tag at the N-terminal.

We have corrected it.

- Line 374: ... at 37°C...

We have corrected it.

- Line 377/378/381: The culture ... / The pellet... / The supernatant ...

We have corrected it.

- Line 382/384: pH 8.0, missing space

We have corrected it.

- Line 387: ... incubated for 1 h and 20 min ...

We have modified it.

- Line 390: ..., and 20 mM imidazole.

We have corrected it.

- The molar ratio of ATP and Mg²⁺ is not stated

It is now fully documented in the new section 4.7 ATP binding

- Line 461: ... the NBDs undergo ...

We have corrected it.

- Line 484-487: to be deleted. These sentences are repeated in lines 488-491

There is now a whole new paragraph describing MD simulations.

- Line 506: Vincristine binding was not presented or mentioned in the results, but in Supp-Figure 25 and therefore, Vincristine binding is not part of the story

We have removed the data corresponding to Vincristine.

- Line 515: which figure shows the mentioned data as a function of ligand concentration, or ATP as the ligand has to be defined

A reference to a figure has been added

- Line 520: the phrase monolayer is not used correctly; a lipid film is created

We have modified it.

- Line 525: ... to ensure a ratio of 1 BmrA dimer per 3000 lipids ...

We have modified it.

- Line 555: ... with 1 μ M ...

We have modified it.

- Line 559-561: repetitions of the phrase bandwidth of 10nm

We have modified it.

- Line 561-562: wrong grammar

We have modified it.

- Line 572: ..., in small increments ..., small is not defined

We have modified it.

- Caption Supp-Figure 13A: sentence is unstructured

We have modified it.

Manuscript number: NCOMMS-24-20516-T

Reviewer #1 (Remarks to the Author):

Overall, the manuscript has improved, and the authors have addressed most of the comments. However, the molecular dynamics (MD) section, including how it was performed, remains unclear to me:

Point ii)

In their rebuttal letter, and upon reviewing Figure 4B, the authors acknowledge that equilibration (from the so-called “production run”) took 250 ns. Therefore, the analyses should be conducted from $t = 250$ ns onwards. I would like to remind the authors that there should be no confusion between thermalization/box equilibration and system equilibration—the latter is system-dependent and should not be used as the basis for conformational analyses. Additionally, the x-axis label in Panel 4D should represent MD simulation time in nanoseconds (ns), not “Frame.”

Upon deposition of the data on Zenodo, we have realized a problem in one simulation resulting from a misleading parameter in the system. To avoid any mistake, we have re-performed the simulations in triplicate. We have redone all the corresponding figures and PCA analysis to match the new data. We slightly modified the text to match the new data, as it strongly backs up the claims initially made and shows a closure of one NBS as hypothesized in the model figure (now Figure 7) and explains cooperativity for ATP binding. We added a new Figure 5 to exemplify this.

We acknowledge the reviewer claim on the system equilibration. At the same time, we are interested in the fast relaxation of the system as it is the part showing most closing. Thus, we performed a 100ns system relaxation first before sending the system into production (detailed in the M&M section), thus removing initial fast relaxation while keeping some large movements to see how the system evolves differently in presence of drug or on its absence.

All figure axes have been changed from frame to nanosecond (ns).

Point iii)

I am disappointed that the authors did not explore allosteric mechanisms through MD simulations, especially since the experimental data presented in this manuscript suggests potential pathways that could be supported by such simulations. However, I understand that this might have required more effort than the authors were able to allocate. Additionally, Reference 37, which was added in response to the review, is not appropriate for ABC transporters.

Thank you for your understanding, we have removed ref 37.

Reviewer #2 (Remarks to the Author):

The authors have addressed all my concerns. I have no further comments.

Reviewer #3 (Remarks to the Author):

The authors have appropriately addressed the raised concerns. I have no further comments.

Reviewer #4 (Remarks to the Author):

Reviewer #5 (Remarks to the Author):
